# Spatio-temporal Patterns of High Mountain Asia's Snowmelt Season Identified with an Automated Snowmelt Detection Algorithm, 1987-2016

Taylor Smith[1], Bodo Bookhagen[1], and Aljoscha Rheinwalt[1]

[1]Institute for Earth and Environmental Sciences, Universität Potsdam, Germany

*Correspondence to:* Taylor Smith (tasmith@uni-potsdam.de)

**Abstract.** High Mountain Asia (HMA) – encompassing the Tibetan Plateau and surrounding mountain ranges – is the primary water source for much of Asia, serving more than a billion downstream users. Many catchments receive the majority of their yearly water budget in the form of snow, which is poorly monitored by sparse in-situ weather networks. Both the timing and volume of snowmelt play critical roles in downstream water provision, as many applications – such as agriculture, drinking-water generation, and hydropower – rely on consistent and predictable snowmelt runoff. Here, we examine passive microwave data across HMA with five sensors (SSMI, SSMIS, AMSR-E, AMSR2, and GPM) from 1987-2016 to track the timing of the snowmelt season – defined here as the time between maximum passive microwave signal separation and snow clearance. We validated our method against climate-model surface temperatures, optical remote-sensing snow-cover data, and a manual control dataset (n=2100, 3 variables at 25 locations over 28 years); our algorithm is generally accurate within 3-5 days. Using the algorithm-generated snowmelt dates, we examine the spatiotemporal patterns of the snowmelt season across HMA. The climatically short (29 year) time series, along with complex inter-annual snowfall variations, makes determining trends in snowmelt dates at a single point difficult. We instead identify trends in snowmelt timing by using hierarchical clustering of the passive microwave data to determine trends in self-similar regions. We make the following four key observations: (1) The end of the snowmelt season is trending almost universally earlier in HMA (negative trends). Changes in the end of the snowmelt season are generally between 2 and 8 days/decade over the 29-year study period (5 - 25 days total). The length of the snowmelt season is thus shrinking in many, though not all, regions of HMA. Some areas exhibit later peak signal separation (positive trends), but with a generally smaller magnitudes than trends in snowmelt end. (2) Areas with long snowmelt periods, such as the Tibetan Plateau, show the strongest compression of the snowmelt season (negative trends). These trends are apparent regardless of the time period over which the regression is performed. (3) While trends averaged over three decades indicate generally earlier snowmelt seasons, data from the last 14 years (2002-2016) exhibit positive trends in many regions, such as parts of the Pamir and Kunlun Shan. Due to the short nature of the time series, it is not clear whether this change is a reversal in a long-term trend or simply inter-annual variability. (4) Some regions with stable or growing glaciers – such as the Karakoram and Kunlun Shan – see slightly later snowmelt seasons and longer snowmelt periods. It is likely that changes in the snowmelt regime of HMA account for some of the observed heterogeneity in glacier response to climate change. While the decadal

increases in regional temperature have in general led to earlier and shortened melt seasons, changes in HMA's crysophere have been spatially and temporally heterogeneous.

## 1 Introduction

More than a billion people across Asia rely directly or indirectly on water sourced from melting snow in High Mountain Asia (HMA) (Bookhagen and Burbank, 2010; Bolch et al., 2012; Kääb et al., 2012; Kang et al., 2010; Immerzeel et al., 2010; Gardner et al., 2013; Hewitt, 2005; Malik et al., 2016). Many catchments receive the majority of their yearly water budget in the form of snow – particularly at high elevations (Barnett et al., 2005). Both the volume of snowfall and the timing of snowmelt play crucial roles in the efficacy of water provision for downstream users, as many applications – such as agriculture and hydropower – rely on consistent and predictable water availability. Many areas also rely on snowmelt to provide a water buffer late in the year when direct precipitation is rare. Any changes in the onset, length, or intensity of the snowmelt season will impact the water security of both high-elevation and downstream communities.

Passive microwave (PM) data has been used to estimate snow depth and snow-water equivalent (SWE) since the launch of the Scanning Multichannel Microwave Radiometer (SMMR) in 1978. Consistent, pseudo-daily measurements became available in 1987 with the launch of the Special Sensor Microwave/Imager (SSMI) series of sensors (Wentz, 2013). PM data is highly sensitive to liquid water present in the snowpack, and is thus a valuable tool for tracking the onset of snowmelt across large, inhospitable, and unmonitored regions. PM data also have the advantage of functioning despite cloud cover, which is ubiquitous in much of HMA during both winter and the Indian Summer Monsoon (ISM) season. Using satellite-derived PM measurements, several authors have tracked the onset, duration, and spatial extent of snowmelt events using a range of approaches including the cross-polarized gradient ratio (XPGR) (Abdalati and Steffen, 1995; Hall et al., 2004), the advanced horizontal range algorithm (Drobot and Anderson, 2001), Gaussian edge detection (Joshi et al., 2001), channel differences (Takala et al., 2003), artificial neural networks (Takala et al., 2008, 2009), diurnal temperature brightness (Tb) variations (Apgar et al., 2007; Monahan and Ramage, 2010; Tedesco, 2007), wavelet-based approaches (Liu et al., 2005), and median filtering of raw PM data (Xiong et al., 2017).

In this study, we adapted a previously published algorithm (Abdalati and Steffen, 1995) that relied on the establishment of a single cutoff threshold for identifying melt phases in Greenland to the more complex and diverse snow regimes of HMA. This algorithm was chosen due to (1) speed of calculation, (2) consistency across the large study area, and (3) reliance on only night-time data, which is less influenced by sporadic daytime melt-refreeze cycles. While previous studies have successfully measured snowmelt in large and homogeneous environments such as Greenland and Antarctica, we found these algorithms as originally formulated ineffective in the highly variable topography and snow dynamics of HMA – particularly when a single passive microwave pixel can encompass several terrain types which may melt at different speeds. Here we present an enhanced and generalized algorithm building on previous work to improve on snowmelt detection in HMA. We then apply this algorithm to PM data from 1987-2016, and use the derived snowmelt dates to examine spatio-temporal snowmelt patterns across the entire HMA region.

## 1.1 Geographic Setting

HMA contains several mountain ranges – the Himalaya, Pamir, Karakoram, Hindu Kush, Tien Shan, and Kunlun Shan – from which flow several large rivers serving more than a dozen countries (Fig. 1). Many of these catchments, such as the Tibetan Plateau, Tarim, Syr Darya, Amu Darya, and Indus, rely on snowmelt for more than 50% of their yearly water budget (Bookhagen and Burbank, 2010; Shrestha et al., 2015). Many communities – particularly those at high elevations or those that depend on surface water for agriculture – are highly reliant on the timing of snowmelt. An early snowmelt season can create a late-season 'water gap' when a dry spell is caused by snow meltwaters disappearing before the start of the next rainy season. These water gaps can also negatively impact flora and fauna which depend heavily on the timing of the appearance of ephemeral water bodies (Bookhagen, 2017). The timing and volume of snowmelt thus has important implications for the environment, direct household water use, agriculture, and hydropower.

Several interacting moisture sources, including the Winter Westerly Disturbances (WWD), Indian Summer Monsoon (ISM), and East Asian Summer Monsoon (EASM), are responsible for the wide range of snowfall regimes across HMA (Fig. 1, inset). The interaction of these climatic regimes with the complex topography of HMA – particularly the vast elevation gradients – creates a diverse set of snowfall regimes (Cannon et al., 2014; Kääb et al., 2012; Immerzeel and Bierkens, 2012; Gardner et al., 2013; Kapnick et al., 2014; Barnett et al., 2005; Dahe et al., 2006; Takala et al., 2011; Cannon et al., 2017).

## 2 Materials and Methods

### 2.1 Datasets

We leverage a combined time series of SSMI (1987-2009), Special Sensor Microwave Imager/Sounder (SSMIS) (2008-2016), Advanced Microwave Scanning Radiometer - Earth Observing System (AMSR-E, 2002-2011), AMSR2 (2012-2016), and Global Precipitation Measurement (GPM, 2014-2016) data, processed to 0.25 decimal degree (dd) resolution by interpolating raw PM swath data at a series of point locations as described in Smith and Bookhagen (2016) (see Supplementary Table S1 for a full dataset listing). In essence, we group all measurements within a $0.125°$ dd radius of each point on a $0.25°$ dd grid and generate a spatially weighted mean value for each swath at that point. The dataset is comprised of 6,399 point locations, with on average 26,000 PM measurements each (long-term average of 2.4 measurements/day for 29 years, with more measurements during the 2002-2016 period).

PM measurements are converted to snow-water equivalent (SWE) using the Chang equation (Eq. 1) (Chang et al., 1987), with modifications for non-SSMI platforms as proposed by Armstrong and Brodzik (2001), and a constant snow density of $0.24$ g/cm$^3$ as proposed by Takala et al. (2011).

$$SD[cm] = 1.59[cm/K] * (Tb_{18V} - Tb_{36V})[K] \tag{1}$$

Studies have noted that SWE estimates from the Chang equation have high uncertainties (e.g., Kelly et al., 2003; Kelly, 2009; Tedesco and Narvekar, 2010; Daly et al., 2012), particularly in dense forests. However, as much of our study area is non-forested – and we use SWE only as a rough estimate of snow volume – we choose to rely on the simple Chang equation rather than a more complex algorithm for SWE estimation.

As control data, we analyze Moderate Resolution Imaging Spectroradiometer (MODIS) percentage snow-covered area (product MOD10C1 V6, 2001-2016, (Hall and Riggs, 2016)) and High Asia Refined Analysis (HAR) surface temperature (tsk, 2000-2014, (Maussion et al., 2014)). While these datasets only cover a subset of our study period, they are among the few independent control datasets available across the entire study area.

## 2.2  Snowmelt Tracking Algorithm

The shift from dry snow, which can be physically characterized as snow crystals in an air background, to wet snow, which replaces the air matrix with water, shifts the primary interaction between PM radiation and the snowpack from volumetric (dry snow) to surface (wet snow) scattering. These scattering changes are reflected in the temperature brightness (Tb) data, and allow wet and dry snow to be differentiated, as the transition from dry to wet snow drastically increases the measured Tb – particularly in the scattering ($Tb_{37V}$) channel. The XPGR, as originally described by Abdalati and Steffen (1995), is defined
as:

$$XPGR = (Tb_{19H} - Tb_{37V})/(Tb_{19H} + Tb_{37V}) \tag{2}$$

This algorithm takes advantage of both the channel difference between the $Tb_{19}$ and $Tb_{37}$ GHz channels as well as the depolarization effects of snowmelt, which increases the differences between the horizontally and vertically polarized channels (Abdalati and Steffen, 1995). In the original application of the XPGR on the Greenland Ice Sheet, a static value of -0.025
was shown to indicate the presence of liquid water in the snowpack, and hence used to separate the year into melting and non-melting phases (Abdalati and Steffen, 1995). We find that for the majority of HMA, the -0.025 threshold is not effective in identifying the onset of snowmelt, as the XPGR-snowmelt relationship is highly variable through time and space.

We modify the XPGR method here to track maximum passive microwave signal separation – or the yearly maximum XPGR, referred to from here on as MXPGR. As the context of seasonal snowmelt in HMA is quite different from that of Greenland,
and sufficiently long-term and spatially diverse in-situ snowmelt data are lacking, we use the MXPGR as a proxy for snowmelt onset to track changes in the snowmelt season year-over-year. Thus, while we do not use the classical literature definition of snowmelt – presence of liquid water in the snowpack – we track a consistent metric related to physical snowpack changes that can be broadly interpreted as the onset of the snowmelt season.

However, the MXPGR is not effective for tracking the cessation of snowmelt. To track the end of snowmelt, we leverage
two additional datasets: (1) the raw $Tb_{37V}$ time series, which rapidly increases as snowpack thins, and (2) a SWE time series calculated from the $Tb_{19}$ and $Tb_{37}$ GHz channels (Chang et al., 1987; Kelly et al., 2003; Tedesco et al., 2015; Smith and Bookhagen, 2016).

We first use a simple peak-finding algorithm, which identifies peaks as points which are larger than their two neighboring samples, to generate a list of potential peaks in the XPGR data. Next, we take the average XPGR value within $\pm 2$ days of each peak to determine not only the simple yearly maximum XPGR, but the highest and temporally widest peak in our XPGR data – termed here the MXPGR. We flag years which have multiple strong and temporally distinct XPGR peaks as unconstrained for snowmelt onset estimation, as the algorithm has trouble consistently identifying the MXPGR in these cases.

To determine the end of the snowmelt season, we choose either the date of the yearly maximum $Tb_{37V}$ value, which corresponds to the thinnest snowpack or to a 'bare earth' signal, or the first date where 4 out of 5 days have been within 2 cm of the yearly SWE minimum. We choose the yearly SWE minimum instead of zero as our SWE threshold for snow clearance because some regions in HMA have a defined melt season but rarely reach zero SWE. This also helps control for uncertainty in shallow SWE measurements, as detecting shallow snow (<5 cm) with PM data is still difficult (Kelly et al., 2003; Armstrong and Brodzik, 2001). A full description of our melt detection algorithm is available in the Supplement (Figs. S1-4).

## 2.3 Manual Control Dataset Generation

Unfortunately, large-scale and several-decades long snowmelt onset and end date records are not available for HMA. Instead, we use HAR (Maussion et al., 2014) and MODIS (Hall and Riggs, 2016) data alongside a manually generated set of control dates for the snowmelt season, determined from the SWE, XPGR, and $Tb_{37V}$ signals by the researchers. We visually identified major peaks (MXPGR), as well as the cessation of snowmelt, by inspection of the time series. We chose a random sample of 25 point locations across our study area, and identify snowmelt dates for each year of the time series (n=1400). We use the calculated length of the snowmelt period (days between the MXPGR and the end of the snowmelt season) as an additional control variable (n=700).

## 2.4 Hierarchical Clustering

Hierarchical clustering is a method used to correlate time series data by intrinsic similarity (Corpet, 1988; Johnson, 1967; Jain et al., 1999; Murtagh and Contreras, 2012; Rheinwalt et al., 2015), which has been used extensively in the environmental research community. We generate clusters from those time series which share the most temporal overlap, or where the periodicity of Tb values have the largest coherence, regardless of their spatial correlation.

We choose the XPGR time series as our clustering variable, as the XPGR is the most sensitive to melt dynamics, integrates multiple Tb frequencies, and is not sensitive to SWE calibration issues. To improve the robustness of our clustering, we combine the disparate single-instrument PM time series into a single coherent time series which leverages the full temporal extent of each dataset (1987-2016), using the following three steps: (1) We standardize the PM signals of the suite of instruments used in this study to a single set of dates, artificially created at daily resolution from the minimum and maximum dates across all satellite datasets, by resampling all individual satellite time series to a daily time step and dropping dates without data. (2) We homogenize the disparate PM time series based on the overlapping portions of individual satellite time series, using linear regression. The results of these regressions can be seen in Tables S2-5, with an example regression at a single point shown in

Figure 2. (3) In order to reduce noise in our cluster analysis, we resample our merged XPGR time series to a 5-day temporal resolution (pentad).

Next, we normalize each merged pentad time series (1987-2016) to a Gaussian distribution, using a percentile mapping approach (Rheinwalt, 2017). We then estimate the Pearson correlation coefficient to classify regions of self-similarity in our XPGR time series (Rheinwalt, 2016). This method computes a Pearson correlation coefficient between each time series, and based on the resulting correlation matrix, computes a set of linkages using the angle between time series in vector space (Murtagh and Contreras, 2012). We use the maximum distance (complete linkage) to split the linkage matrix, which is favorable because it ensures a minimum intra-cluster correlation. An average linkage scheme was tested and produced heterogeneous cluster sizes with outliers. We choose our cluster threshold from the hierarchical clustering dendrogram (Fig. S6), which maximizes cluster size while minimizing cluster internal diversity (Fig. S7). We emphasize that the correlation is based on the temporal co-evolution of the time series, and is less sensitive to the relative magnitudes of peaks and troughs between data points. For an oscillating time series, the magnitude of the Pearson correlation coefficient is driven by the synchronization of peak timing, especially in normalized time series. The combination of several sensors in this study may impact the magnitudes of the resultant time series, but will not have an outsized effect on the timing, and thus clustering, of our time series.

# 3   Results

## 3.1   Melt Algorithm Validation

### 3.1.1   Comparison with Manual Control Dataset

The agreement between manually clicked snowmelt dates and algorithm-derived snowmelt dates is generally within 3 days, with 70% or more of MXGPR and snowmelt end dates falling within 5 days of the control dataset (Table 1). We find the lowest standard deviation for the end of melt, which is to be expected given that the end of snowmelt is determined by both snow clearance and the $Tb_{37V}$ signal, and thus is more tightly constrained than the MXPGR date. The MXPGR date, while having a low average offset, has a high standard deviation as the algorithm sometimes has trouble correctly choosing the MXPGR when a snow season has several large storms, or several periods of melting and refreezing (cf. Fig. 3). Thus, errors in identification of MXPGR will naturally have a higher standard deviation due to the presence of more relatively large misclassification errors.

Diverse snow seasons are shown from an example location (71.25E, 36.75N), over six years of data (Fig. 3). Despite clear inter-annual variations in the temporal distribution of SWE, there exists high correlation between the algorithm-derived melt dates and our manually chosen melt dates. In the sample data, the first and third snow seasons have multiple peaks which could possibly be related to the true onset of the snowmelt season, and these years are flagged as unconstrained. The second and fourth years of data have a simple structure with a well-defined peak and a pseudo-linear melt during the spring season. The fifth year of data has a strong late-season XPGR peak, implying that there was significant snow buildup after an initial early season XPGR peak and melt phase. The last year of data shows a mismatch between the algorithm and control datasets, where it is difficult to determine the best candidate for the MXPGR. The algorithm picks the wider XPGR peak (earlier in the season),

while we chose the thin but high peak later in the season as more closely following the end of snow buildup. Across all years of data shown here, the snowmelt end date is well matched between the algorithm and manual datasets.

### 3.1.2 Comparison with MODIS Snowcover Data

The MODIS sensor onboard Terra (product MOD10C1 V006, (Hall and Riggs, 2016)) provides an additional estimate of
snowcover from an optical, instead of PM, instrument. While MODIS cannot provide accurate measurements of fractional snow covered area (SCA) in the presence of clouds, it represents an independent control on the snowmelt end date (Fig. 4). In Figure 4A, the MODIS snow-clearance date is defined as the first day when five out of seven days have less than 5% SCA, and the data are cloud free. Only those dates where there is no cloud cover within seven days of the end of the snowmelt season are used in Figure 4B, which illustrates the consistently low SCA fraction at our algorithm-derived end of the snowmelt season.
While the agreement between algorithm and MODIS snowmelt end days is generally high (cf. Fig. 4A), there remain significant outliers. It is likely that some larger outliers are due to poorly flagged clouds in the MODIS dataset (cf. Fig. S2). We rely here on the MOD10C1 product, as other snowcover products such as NOAA Global Multisensor Automated Snow and Ice Mapping System (Romanov, 2017) utilize a combination of optical and passive microwave data, and thus do not represent a truly independent control dataset.

### 3.1.3 Comparison with HAR Surface Temperature Data

HAR provides surface temperature at hourly intervals from 2000-2014 at 30 km spatial resolution over the entire study area (Maussion et al., 2014). Using this data, we derive (1) the full-day average surface temperature, (2) the average daytime surface temperature, and (3) the daily surface temperature range at each MXPGR date (Fig. 5).

While the relationship between surface temperature and MXPGR isn't as clearly defined as the comparison between MODIS
SCA and snowmelt end, the highly variable surface temperature and positive daytime temperatures at the MXPGR dates imply that the MXPGR is likely linked to melt-refreeze cycles, snowpack metamorphism, or the presence of liquid water in the snowpack.

### 3.2 Application: Spatial Patterns of Snowmelt Period

We apply our algorithm on a pixel-by-pixel and year-by-year basis to identify the onset of the snowmelt season – here proxied
by the MXPGR – as well as the end of the snowmelt season. We also use the number of days between the MXPGR and the end of the snowmelt season to calculate the snowmelt period for each year. The long-term average snowmelt period is shown in Figure 6.

The length of the snowmelt season varies significantly across HMA (Fig. 7). In many low-elevation areas, such as the Ganges Plain, and low-SWE areas, such as the central Tarim Basin, the snowmelt period is very short. Higher-elevation zones, and in
particular the Tibetan Plateau, see snowmelt periods of several months. While both elevation and the amount of SWE impact snowmelt, these are not the sole determinants of the length of the snowmelt season (Fig. 7). Each of the major catchments

(cf. Fig. 1) has a unique MXPGR, snowmelt end date, and snowmelt-period distribution, based on the various climate and topographic forcings present in each catchment.

## 3.3 Hierarchical Clusters

Cluster selection criteria can be seen in Figures S6-7. We choose our dendrogram cutoff (distance threshold in vector space) based on a combination of the number of generated clusters, the internal variation within those clusters, and the average resultant cluster size. In our case, we choose a distance cutoff of 1 radian from the complete linkage matrix (minimum intra-cluster correlation 0.525), which results in 285 clusters (Fig. 8).

While the hierarchical clusters are not based on any explicit spatial relationships, many of the clusters fall into spatially coherent groups. For example, the Pamir Knot and Tarim Basin both form large, coherent clusters based on the similarity of their snowfall and snowmelt patterns. The large number of small clusters throughout the Himalaya indicate that the region is not climatically uniform, and small-scale variations in topography and climate have strong impacts on the snowmelt regime.

## 4 Discussion

### 4.1 Spatial Melt Patterns from Hierarchical Clustering

As can be seen in Figure 3, there exists significant inter-annual variation in the timing the snowmelt season. This is particularly true of areas impacted by the WWD, which often have multiple snowfall events starting in winter and lasting until spring (Cannon et al., 2014). As one year may receive a small late season storm, and thus see a maximum in the spring, while the next year may receive a large summer storm, and thus peak in the summer, analyzing trends at a single point in space is difficult.

To mitigate the influence of inter-annual variation in determining long-term trends in the timing of the snowmelt season, we group our data into self-similar clusters using hierarchical clustering. We do not filter our generated clusters based on size or self-similarity, as we do not use our clusters to generate a single averaged or representative time series for each cluster, as is often done in climate analyses. Due to inter-annual variations in SWE and the timing of the snowmelt season, fitting a linear regression through only 29 years of data does not provide statistically significant results for the majority of HMA. Instead, we use our clusters to group sets of algorithmically-derived snowmelt dates, and fit linear models on a cluster-by-cluster basis. By leveraging the snowmelt dates of a set of time series in parallel, we are able to identify statistically significant changes in the timing of the snowmelt season, as well as changes in the length of the snowmelt period (Fig. 9). To reduce noise from low-SWE and very short snowmelt period areas, we remove areas from the subsequent analyses with long-term average melt periods of less than 20 days. We also remove MXPGR dates that are flagged as unconstrained (when there are multiple candidate dates) to limit the impact of unreliable data on our analysis.

MXPGR is trending earlier (negative trend) in HMA outside of a small band running from the Karakoram through the interior Tibetan Plateau (Fig. 9A). In another snowmelt study leveraging SSMI and QuickSCAT data in HMA, Xiong et al. (2017) find a similar distribution of positive and negative snowmelt onset trends. However, a direct comparison with their

results is difficult due to differences in the temporal and spatial resolution of source data, filtering methods, and statistical treatment of SWE trends. Negative snowmelt onset trends have also been previously observed in Central Asia (Lioubimtseva and Henebry, 2009; Dietz et al., 2014), the Himalaya (Lau et al., 2010; Panday et al., 2011), and the Tibetan Plateau (Xu et al., 2017).

A complex pattern of regionally increasing and decreasing spring snow depth in the Tibetan Plateau has been observed since the 1970s (Zhang et al., 2004; Che et al., 2008; Wang et al., 2013), which could help account for the mixed MXPGR trends observed in the Tibetan Interior. High-elevation zones in the upper Indus catchment, running from the Karakorum in a south-eastward direction, have seen increased precipitation over the past decades due to increases in the strength of the WWD (Cannon et al., 2015; Norris et al., 2016; Treydte et al., 2006).

Temperatures in HMA are increasing faster than the global average (Vaughan et al., 2013; Lau et al., 2010), and are likely the primary driver of the almost universal earlier snowmelt end dates as seen in Figure 9B. Increased temperatures have likely both reduced overall SWE amounts, by causing more precipitation to fall as rain, and decreased SWE persistence into the spring and summer months. These changes have helped drive a 2-8 day/decade earlier end to the snowmelt season (Fig. 9B).

The length of the snowmelt season is shortening in much of HMA, with the exception of small areas in the Pamir, Tien Shan,
and Karakoram regions (Fig. 9C). We attribute this to a combination of increased WWD storm intensity, and increases in late season storms, which could help extend the snowmelt season slightly later into the year (Cannon et al., 2016; Norris et al., 2015; Kapnick et al., 2014). In general, however, the snowmelt season is shortening throughout HMA. Intensification of the spring runoff regime in HMA has been observed in both model (Lutz et al., 2014) and empirical (Dietz et al., 2014; Bookhagen and Burbank, 2010; Stewart, 2009) data.

**4.2   Temporal Heterogeneity in Snowmelt Trends**

Not only are changes in the snowmelt regime spatially complex (e.g., Fig. 9), but they exhibit distinct temporal heterogeneity as well.

Changes in MXPGR do not have a bias towards early or late onset snow regimes (Fig. 10A). The end of the snowmelt season is almost universally negative (earlier), excepting a few isolated areas in the Kunlun Shan (cf. Fig. 9). The majority of locations
show negative (shorter) trends in snowmelt period. Strong negative changes in the snowmelt period are biased towards areas with long melt seasons (120 days or more). This implies that high-elevation areas, such as the Tibetan Plateau, and high-SWE areas, such as the Karakoram, will see a relatively stronger compression in the length of the snowmelt season. While changes in the MXPGR date are partially responsible, the main driver of shorter snowmelt periods is the earlier end of the snowmelt season across most of HMA.

Several-decade long trends conceal short-term fluctuations in the snowmelt regime of HMA. To assess the impact of the analysis timeframe on our regression results, we analyzed trends with window sizes ranging from four years to 28 years, across all possible start-year and window-size combinations, averaged over the entire study area (Fig. 11).

Trends are universally negative for the MXPGR and the end of the snowmelt season, as well as for the snowmelt period, between 1988 and 1995, regardless of the timeframe over which the regression is performed. While there were some short

positive trends in snowmelt end date (5-10 years) starting in the mid 1990s, trends in end dates and snowmelt period are generally negative. Although long-term trends in MXPGR date (longer than 20 years) are negative, recent trends (after 2002) are positive when considered at timeframes of 5-10 years. This implies that while the three-decade trend in MXPGR dates has been negative, the trend has become more variable in the past decade.

It is clear that decadal trends (cf. Fig. 9) are not consistent throughout the entire study period (cf. Fig. 11). When trends in the first half (1988-2002) and second half (2002-2016) of the data are compared, distinct regional patterns are apparent (Fig. 12).

    The lack of statistically significant trends limits some interpretations, particularly with regards to changes in the snowmelt period. Nowhere in HMA are MXPGR trends consistent in both analysis periods. While many snowmelt end dates have

remained negative in both time periods, trends in parts of the Pamir and Karakoram have moved from negative to positive, and those in the Tien Shan have become less negative (cf. Fig. S8). A similar story is apparent when MXPGR dates are considered, where the Tien Shan and parts of the Pamir have moved from negative to positive trends. Unfortunately, due the the climatically short nature of the dataset, it is not clear whether this change represents inter-annual variability or a reversal of a long-term trend.

**4.3   Hydrologic Implications**

The spatially and topographically complex changes in MXPGR, snowmelt end, and snowmelt period make interpretation of downstream impacts difficult. The long-term trend in HMA of a shortened and earlier melt season will impact downstream populations who rely on the consistent timing and volume of spring and summer runoff (Archer and Fowler, 2004; Barnett et al., 2005). Already the impacts of precipitation intensification and shifts in the snowmelt season have been felt in many

regions (Barnett et al., 2005; Stewart, 2009). These trends are likely to continue as temperatures rise across HMA, and each major catchment will feel the impacts of a shortened snowmelt season, regardless of changes in the start and end dates of melt.

    Many regions rely on glaciers as their only water source between the end of snowmelt and the beginning of major precipitation systems (Bolch et al., 2012). This important water reserve is certain to be impacted by, and reflect changes in, the snowmelt regime of HMA, as the timing of precipitation has been shown to be an important factor in the response of glaciers to climate

change (Maussion et al., 2014; Wang et al., 2017). While many regions have seen rapid glacier retreat (Bolch et al., 2012; Kääb et al., 2012, 2015; Scherler et al., 2011) there exist regions of glacier stability and even growth, such as the Karakoram (Hewitt, 2005; Gardelle et al., 2012) and Kunlun Shan (Gardner et al., 2013; Yao et al., 2012). Our results (cf. Fig. 9) show longer snowmelt periods in parts of the Pamir, later snowmelt end dates in parts of the Karakoram and Kunlun Shan, and relatively less negative trends in snowmelt end in the Pamir when compared with the rest of HMA. These regions overlap with both the

'Karakoram Anomaly' and positive glacier mass balances in parts of the Kunlun Shan and Pamir, implying that changes in the timing of the snowmelt season could be partially responsible for regional heterogeneity in glacier change.

    The majority of HMA, however, exhibits a three-decade long trend towards an earlier end of the snowmelt season. Earlier snow clearance increases the absorption of solar radiation, and thus stores more heat at high elevations and generates a positive feedback (Willis et al., 2002). As seasonal snow is removed earlier from glacier regions, glacier melt will accelerate. In general,

glaciers in HMA are decreasing in volume and shrinking, which fits with the observed long-term decrease in snowmelt end dates (cf. Figs. 9, 10, 11), despite clear spatial and temporal heterogeneity in these trends (cf. Fig. 12).

## 4.4 Caveats of the Method

Our algorithm-derived snowmelt end dates and those derived from the independent MOD10C1 product show close alignment, indicating that the algorithm is well-suited to identifying the end of the snowmelt season (cf. Fig. 4). The identification and interpretation of the MXPGR, however, is more difficult. While previous work has used the XPGR to identify the presence of liquid water in snowpack, this relationship has not been confirmed with in-situ data in HMA. While it is likely that XPGR peaks are linked to melt-refreeze cycles, liquid water, or other snowpack metamorphism, these conclusions lack true in-situ controls. This uncertainty, combined with the periods of melt and refreeze and late-season storms in much of HMA, make linking MXPGR to the onset of the snowmelt season difficult. The multiple peaks and troughs in the XPGR data also hamper the identification of a single strong peak to classify as the MXPGR. Without rigorous measurements of surface air temperature or in-situ monitoring of snowmelt, the efficacy of our algorithm for identifying snowmelt onset cannot be directly confirmed.

Despite these drawbacks, MXPGR dates are correlated with the day of year that HAR temperatures first start to increase and MODIS SCA is maximal (Fig. S2), and provide a single consistent proxy for the onset of the snowmelt season in this vast and largely unmonitored area. Furthermore, MXPGR is associated with days with a high temperature range and on-average positive daytime surface temperatures (cf. Fig. 5). This implies that rising daytime temperatures, in conjunction with solar radiation, are linked to MXPGR dates in our study area. However, as we lack a direct control dataset for snowmelt onset, and there is a high degree of variance in the HAR surface temperature-MXPGR relationship, MXPGR dates and trends therein should be considered as less reliable than trends in snowmelt end dates.

## 5 Conclusions

This study presents a snowmelt tracking algorithm based on the cross-polarized gradient ratio, native passive microwave (PM) signal, and a rough estimate of snow-water equivalent (SWE). We do not rely on static thresholds to classify the snowmelt season across our diverse study region, but instead rely on identifying the snowmelt signal from intrinsic properties of each individual time series. The algorithm leverages passive microwave data from the Special Sensor Microwave/Imager (SSMI), Special Sensor Microwave Imager/Sounder (SSMIS), Advanced Microwave Scanning Radiometer - Earth Observing System (AMSR-E), AMSR2, and Global Precipitation Measurement (GPM) satellites (1987-2016) to track the characteristics of the snowmelt season across High Mountain Asia (HMA). We examine large-scale spatial patterns in the snowmelt regime and identify trends in the timing of snowmelt across HMA over the past three decades using hierarchical clustering.

We find the following four key points: (1) The snowmelt season is ending earlier in much of HMA (negative trend), with magnitudes between 2 and 8 days/decade (5-25 days total over 29 years). The length of the snowmelt season is shortening in the majority of HMA, despite some regions of delayed snowmelt onset. (2) Negative changes to the end of the snowmelt season are felt most strongly in areas with long snowmelt seasons (as averaged over three decades), such as the Tibetan Plateau

and high-SWE areas in the Himalaya, Karakoram, and Tien Shan. (3) While three-decade long trends indicate earlier end dates for the snowmelt season, recent (2002-2016) trends are positive (later snowmelt end dates) in many regions of HMA. These changes could be due to inter-annual variability or a reversal in the long-term trend. (4) Areas with slightly longer snowmelt seasons or later MXPGR dates overlap with regions of positive glacier mass balance, such as the Pamir and Kunlun Shan. This implies that changes to the snowmelt regime of HMA could help account for some of the observed regional glacier changes. In general, however, regional warming has led to earlier and shortened melt seasons in much of HMA. These changes are spatially and temporally complex, and will require further local and high-spatial resolution assessments to fully understand changes in HMA's cryosphere.

*Code availability.* The code used in this study is available online at: https://github.com/UP-RS-ESP/SnowmeltTracking

*Author contributions.* T.S. and B.B. designed the study, T.S. prepared and analyzed the PM data. B.B. and A.R. contributed to the development of the methodology. T.S. wrote the manuscript with input from all authors.

*Competing interests.* The authors declare that they have no conflict of interest.

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

**Table 1.** Summary statistics comparing manual control dataset and algorithm dataset (n=2100, 28 snowmelt seasons at 25 locations).

| Variable | Mean Offset (days) | Mean Absolute Offset (days) | Standard Deviation | RMSE | Percentage of Algorithm Dates Within 3/5/10 Days of Control Dates |
|---|---|---|---|---|---|
| MXPGR | -0.23 | 5.51 | 16.71 | 16.71 | 68 / 80 / 90 % |
| Snowmelt End | -1.3 | 5.0 | 9.74 | 9.82 | 49 / 70 / 89 % |
| Snowmelt Period | -0.25 | 7.44 | 16.1 | 16.1 | 47 / 64 / 82 % |

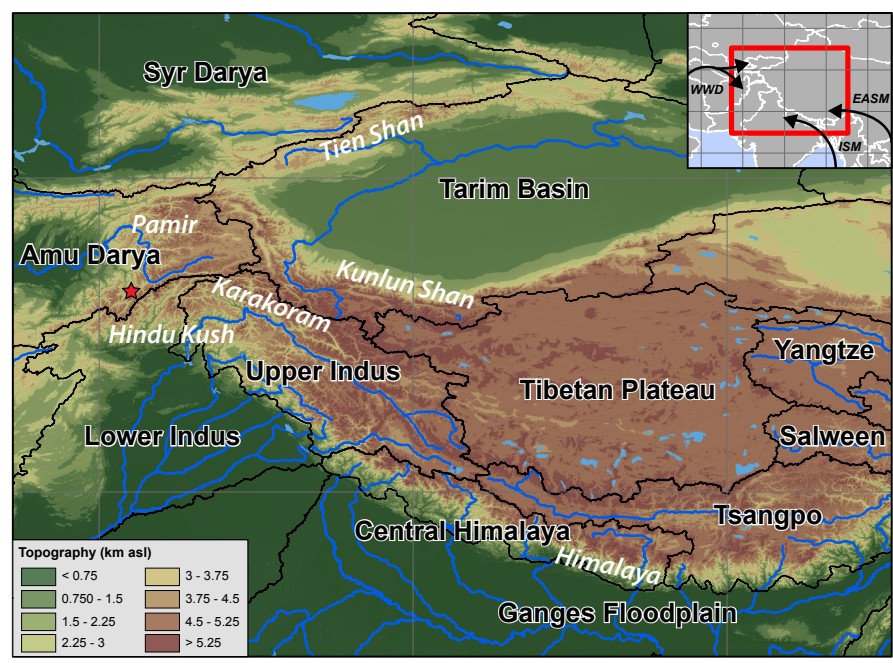

**Figure 1.** Topographic map of the study area across High Mountain Asia (HMA), with major catchment boundaries (black lines and labels in black font with white border) and major mountain ranges (white font). Inset map shows wind direction of major Asian weather systems (WWD: Winter Westerly Disturbances, ISM: Indian Summer Monsoon, EASM: East Asian Summer Monsoon) on top of political boundaries. Red star indicates the location used for Figures 2 and 3.

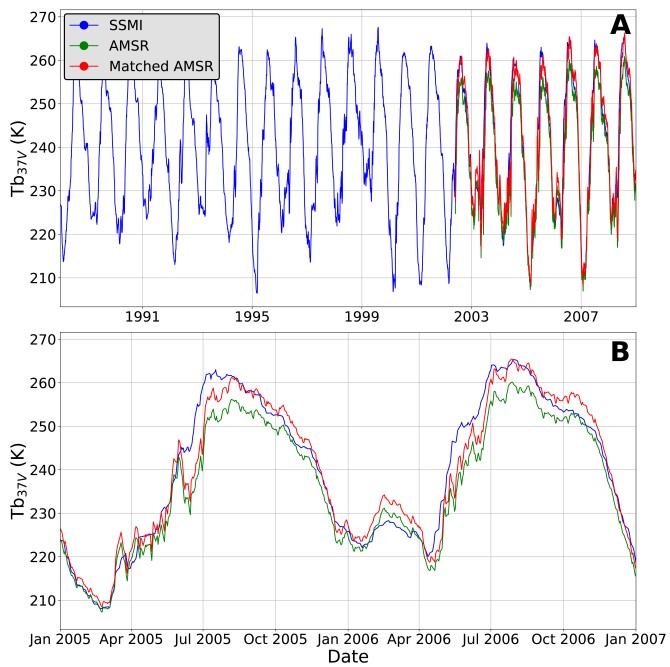

**Figure 2.** (A) Sample time series showing SSMI (blue) and AMSR-E (green) Tb$_{37V}$ frequencies, with linearly matched modified AMSR-E Tb (red), 1987-2009. Data taken from 71.25E, 36.75N (cf. Fig. 1). (B) The same data as panel A but for two seasons (2005-2007).

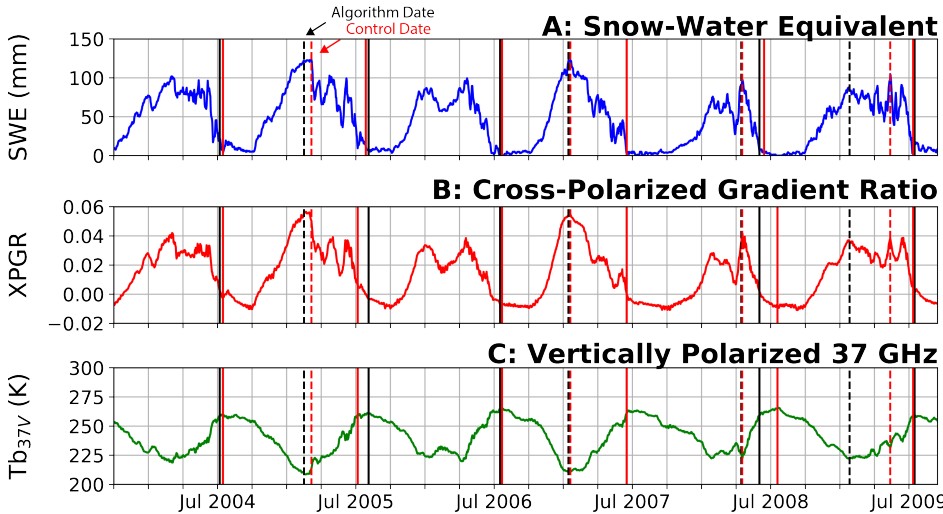

**Figure 3.** Sample data from 71.25E, 36.75N (cf. Fig. 1) showing: (A) Snow-Water Equivalent (SWE) based on the Chang algorithm (Chang et al., 1987), (B) Cross-Polarized Gradient Ratio (XPGR), and (C) vertically polarized temperature brightness at 37 GHz ($Tb_{37V}$) measurements. MXPGR (dashed lines) and end of melt (solid lines) are black for algorithm dates, and red for control dates. Lack of red lines indicates temporal overlap of algorithm and control dates. Years with multiple distinct peaks (e.g., 2004, 2006) are flagged as unconstrained, and not used for further analysis.

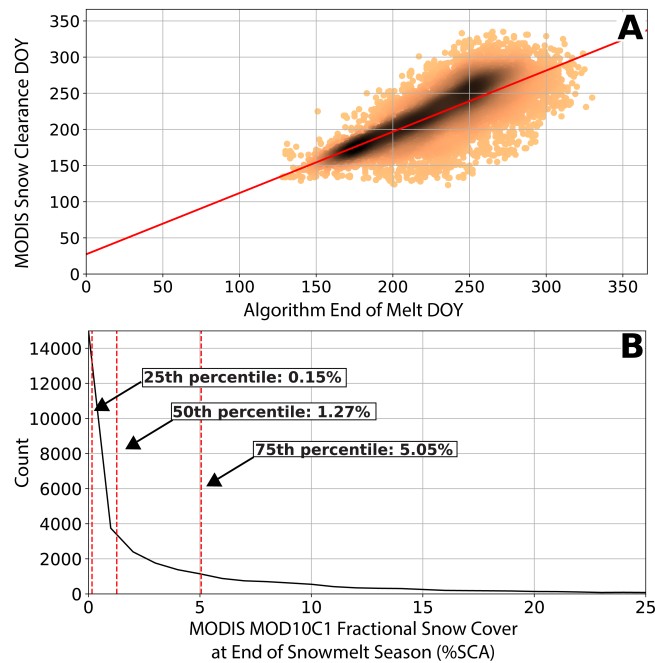

**Figure 4.** (A) Comparison of MODIS MOD10C1 (Hall and Riggs, 2016) and algorithm-derived end of the snowmelt season days of year, with darker areas indicating high point densities. We find strong agreement between the snowmelt end dates derived from both datasets (slope = 0.85, $R^2$ = 0.58, n = 34,468), despite the presence of outliers. (B) MODIS snow covered area fraction at the algorithm-derived end of the snowmelt season. This shows, for example, that over all algorithm-determined snowmelt end dates, the median SCA was 1.27%, and that SCA is below 5% in the majority of cases. Areas with snowmelt periods of less than 20 days are removed from this analysis.

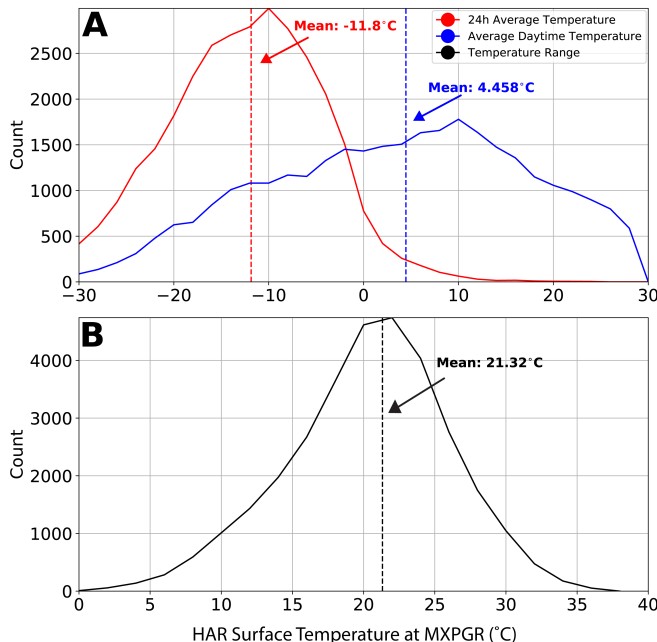

**Figure 5.** (A) HAR full-day average surface temperature (red), daytime average surface temperature (blue), and (B) daily surface temperature range (black) at the algorithm-derived MXPGR date (n = 31,583). Full-day and daytime average temperatures show distinctly different distributions, with full-day temperatures averaging below $0°$C and daytime temperatures above. This relationship, as well as the large daily temperature range, imply that the algorithm-derived MXPGR dates occur at or near the transition from sub-freezing to above-freezing temperatures.

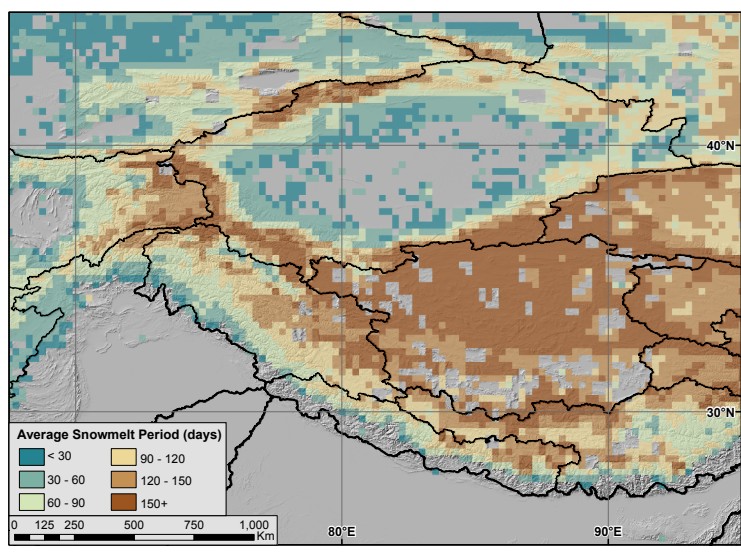

**Figure 6.** Average snowmelt period across HMA from 1987-2016. Snowmelt period ranges from less than a month to several months, depending on geographic location, elevation, and local and regional climatic conditions. Locations with long-term average snowmelt periods less than 20 days are removed. Topographic hillshade in background. Grey areas indicate water bodies, low-SWE areas, and short snowmelt period areas that are excluded from the analysis.

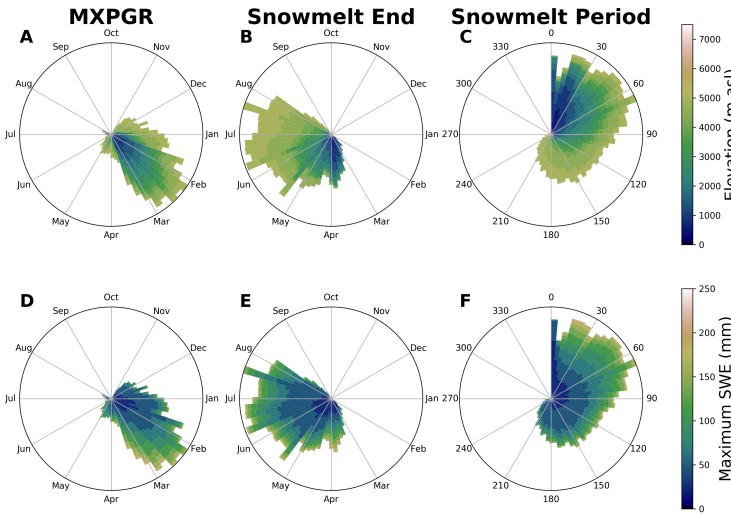

**Figure 7.** MXPGR (A,D), snowmelt end (B,E), and snowmelt period (C,F) for the entire study area, colored by elevation (A-C) and snow depth (D-F) bins. Radial bin heights (radial distance from the center) indicate relative number of pixels at each day of year (i.e. area). While very short snowmelt periods show a distinct low-elevation, low-SWE bias, in general melt onset and end dates are well distributed throughout elevation zones and SWE amounts.

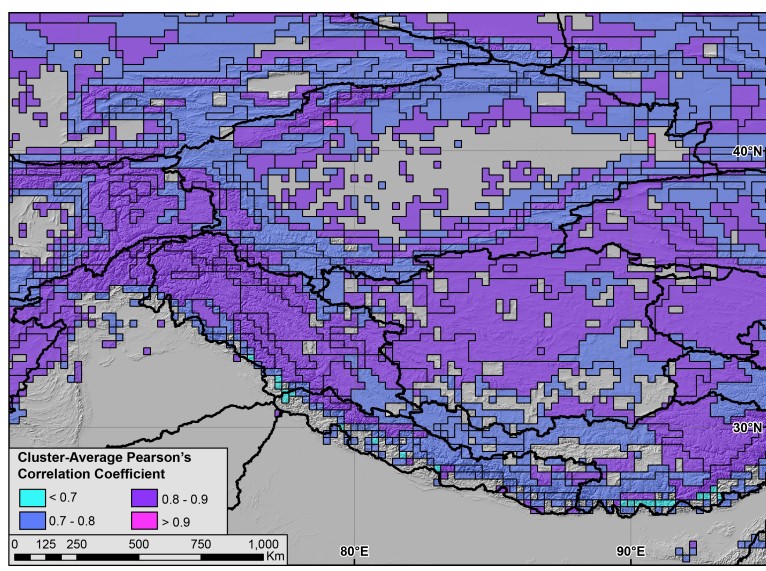

**Figure 8.** Hierarchical clusters (black outlines), as determined from the rank-order correlation coefficients of the 5-day resampled, merged, and linearly matched XPGR data (1987-2016). Colors indicate cluster-average internal diversity (average Pearson's correlation coefficient between members in the same cluster). Grey areas indicate water bodies, low-SWE areas excluded from the analysis, or areas with irregular PM signals which fail to cluster.

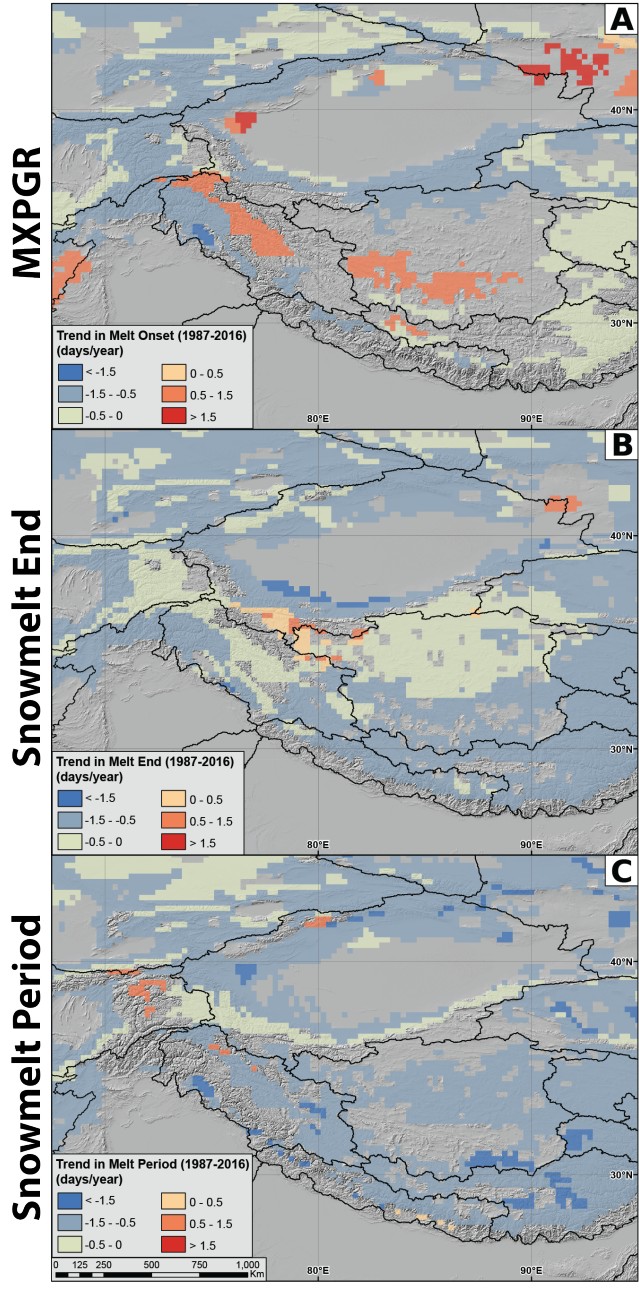

**Figure 9.** Significant (*p* <0.05 ) trends in date of (A) MXPGR, (B) snowmelt end, and (C) snowmelt period, 1987-2016 for the cluster areas (cf. Fig. 8). The MXPGR is generally moving earlier outside of the Tibetan Plateau-Karakoram region, and moving slightly later in a high-elevation zone running from the Karakoram through the Tibetan Plateau interior, as well as parts of the Himalaya. The end of the melt season is moving earlier in the vast majority of HMA, at varying rates. Consequently, snowmelt period is also shrinking in much of HMA, with the exception of small parts of the Pamir, Karakoram, and Tien Shan.

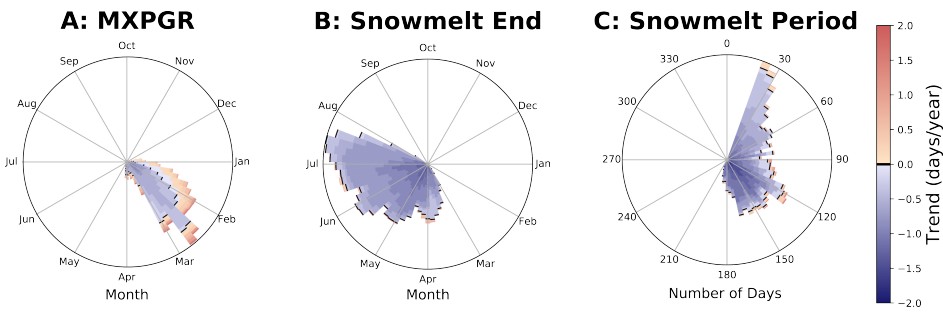

**Figure 10.** 29-year average (A) MXPGR, (B) snowmelt end, and (C) snowmelt period, colored by trend (1987-2016), with radial bin heights (radial distance from the center) indicating relative number of pixels (i.e. area) at each day of year. Black lines indicate zero trend. Data taken only from areas with statistically significant trends ($p <0.05$, cf. Fig 9). Changes in snowmelt end date are positive in very few areas. Negative changes in snowmelt period (shortening) are relatively larger in long snowmelt-period areas.

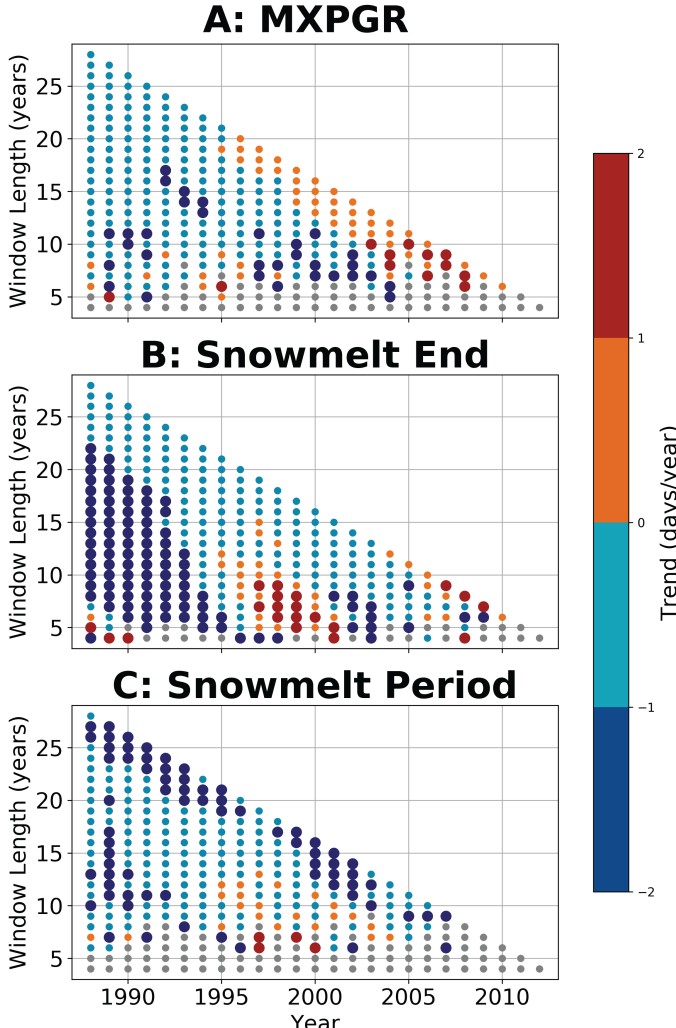

**Figure 11.** Impact of window length on measured trends in (A) MXPGR, (B) snowmelt end, and (C) snowmelt period over the entire study area. Each dot represents trends over a single window size (4 to 28 year) and start year (1988-2012) combination. Regressions are performed using the same clusters as shown in Figure 8. Only statistically significant trends ($p < 0.05$) are included in this analysis; gray dots indicate lack of significant trend. Larger dots indicate positive or negative trends larger than 1 day per year. Trends in snowmelt period and snowmelt end dates are generally negative regardless of which years the trend is assessed over, excepting short periods in the late 1990s and 2000s. MXPGR dates are positive over short time periods starting in the late 1990s, and negative over earlier time periods and longer time windows.

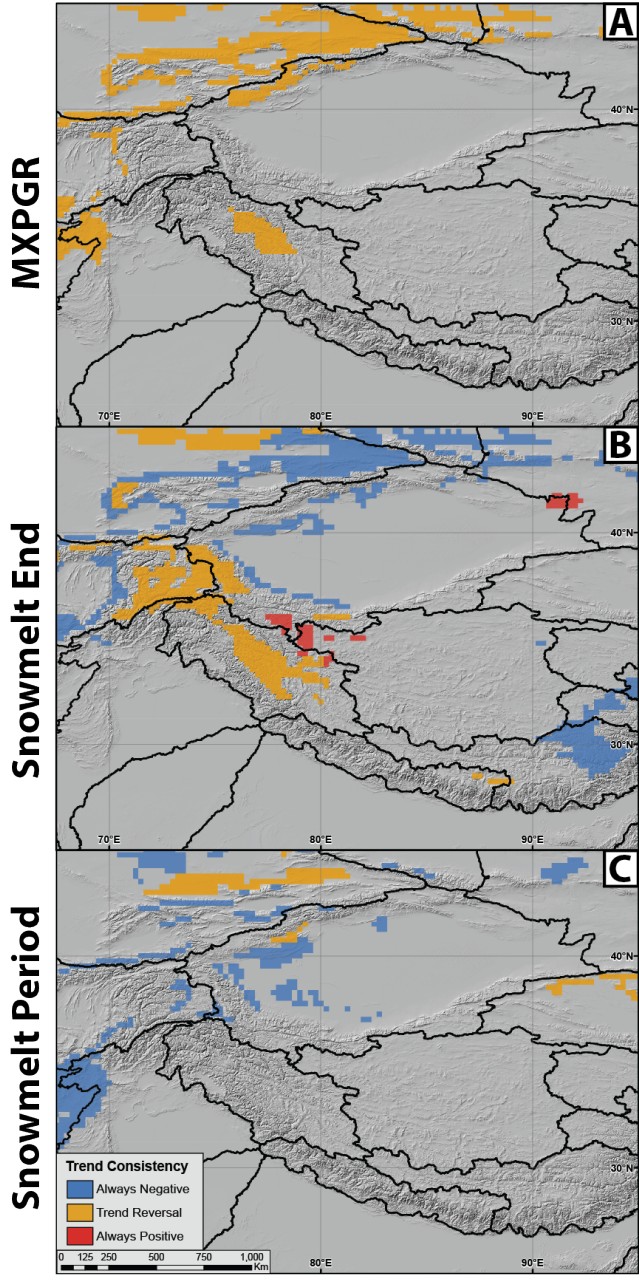

**Figure 12.** Impact of analysis period (1988-2002 or 2002-2016) on measured trends in (A) MXPGR, (B) snowmelt end, and (C) snowmelt period. Grey areas indicate lack of statistically significant ($p <0.05$) trends at one or both analysis periods. Much of HMA lacks significant shorter-term trends in MXPGR and snowmelt period, highlighting the complexities and inter-annual variation in the snowmelt season. While northern HMA has maintained a negative trend in snowmelt end throughout both analysis timeframes, a large region running from the Pamir east has had a reversed trend from negative to positive in the last decade. Regression results at both individual timeframes are available in the Supplement (Fig. S8).