# Peer review of "Spatio-temporal Patterns of High Mountain Asia's Snowmelt Season Identified with an Automated Snowmelt Detection Algorithm, 1987-2016"

_The Cryosphere, 2017_

## Referee Comment (RC1) · Anonymous Referee #1 · 15 Jun 2017

In this paper, the authors use passive microwave data to identify snowmelt onset, snowmelt end and snowmelt period over the High Mountain Asia (HMA) region. They verify the results of an automated algorithm by comparison to manually identified dates in the microwave signal and find it matches to within 5 days. They then use the algorithm to calculate the melt onset, end and period over 29 years and evaluate trends across the region. The paper is well written and provides a long-term record of snowmelt trends across a region where snowmelt is a critical source of water supply. They use an existing method for identifying melt, but apply new techniques for detecting onset and end, as well as a hierarchical clustering method to identify spatial patterns in the data. This paper contributes to the literature in an understudied area of

the world.

My main feedback is that the lack of validation data for this technique raises a number of questions. It would be useful to see validation of the approach in this region that would lend confidence to the results, independent of the microwave data. Some possible data sources that could be used include snow covered area from MODIS or VIIRS to estimate snowmelt end dates. Discharge data, if available, could be used to verify the onset of melt by evaluating the rising limb of the snowmelt hydrograph. Similarly, it may be possible to examine shifts towards earlier melt timing by looking at the hydrograph centroid. (See Regonda et al. 2004, Seasonal Cycle Shifts in Hydroclimatology over the Western United States, Journal of Climate, Vol. 18). Alternatively, temperature data may provide some verification of onset dates. If these data are not available, then demonstrating the approach in an area with data would be useful.

General comments:

1. The 36 GHz signal saturates out in deep snow, which I expect much of this area experiences. How does that affect the gradient ratio approach, since the difference may remain fairly constant for much of the season? How do you know you're selecting the actual maximum XPGR?

2. Related to question 1, the XPGR seems to follow the calculated SWE signal. How does the calculated SWE compare to general estimates of SWE in the region? Is it reasonable, or is there evidence of signal saturation?

3. It is interesting that some of the trends change after 2002, when several additional instruments begin to be available and are included in the analysis. Is it possible that differences in the sensors are causing different results?

4. Following on question 3, in section 2.3 the method used to merge the datasets for the hierarchical clustering analysis is described. Was this merged dataset also used in the snowmelt tracking analysis? If not, then explain why differences in the sensors

wouldn't impact the estimated melt onset and end dates. If yes then this description should be included earlier.

5. The manual selection of dates based on the time series seems subjective. It would be useful to include additional information on how those dates were selected. For example in Figure 3 – in both 2009 and in 2010 there were two peaks of similar magnitude during the winter season. In 2009 the one closer to the end of the season was selected despite appearing less than the earlier one. In 2010 the one very early in the season was selected despite there being an almost equal peak later on. The description in section 3.1 should be clearer.

Specific comments:

1. Page 3, Line 25: why was this algorithm chosen over the other methods referenced in the introduction?

2. Section 2.1: additional background information on passive microwave detection of snow and snowmelt is needed, specifically on how the signal is affected by liquid water in the snow at different frequencies.

3. Page 4, Line 7: Was SWE calculated using the Chang algorithm on Tb from the different sensors? Or are you using the SWE products developed for the different sensors? Adding the equation would be useful. How do you combine multiple sensors when available?

4. Page 5, line 2: Not sure what is meant by "regularize".

5. Page 5, lines 23-25: How does the standard deviation of the melt onset date vary spatially? It seems this approach would work best in high elevation/deep snow regions, whereas along the edges in lower elevation where the snow is more ephemeral there might be more error. This would also affect estimates of melt period.

6. Page 6, line 8: Where you say, "snow is present for less than a month on average." Are you referring to the snowmelt period or the entire snow season? That sounds like

the entire season, but everywhere else is referring to melt period.

7. Page 6, line 24: You say, "As can be seen in Figure 3, inter-annual variation in snowfall can cause large disparities in the yearly dates of snowmelt onset and end." Based on Figure 3 there doesn't appear to be a lot of variability – the peak SWE is around 100mm each year. Are you referring to timing of snowfall events?

8. Figure 1. Can you identify on the overview map the location of the sample data shown in figures 2 and 3?

9. Figure 4: What do the gray areas on the plateau represent? Provide an explanation, similar to figure 6.

―――――――――――――――――――――――――

---

## Referee Comment (RC2) · Anonymous Referee #2 · 26 Jun 2017

General Comments

Passive microwave satellite data have been used for snowmelt onset detection on nearly all components of the cryosphere (e.g. ice sheets, sea ice, lake ice, and seasonal snow cover) mainly at the northern middle to high latitudes and Antarctica, where there is permanent or relatively stable snow accumulation each winter. It has also been used to classify daily freeze/thaw state dynamics for global vegetated land cover regions without distinguishing individual elements of the landscape (e.g. soil, vegetation, snow). The current study detects snowmelt onset/end dates and snowmelt season in Tibetan Plateau and its surrounding mountain ranges, and analyzes trends

in snowmelt timing during the 1987-2016 period. Snowmelt onset detection is based on the crossed-polarized gradient ratio (XPGR) algorithm developed in previous studies, and the end of snowmelt is determined by time series of Tb37v and snow water equivalent (SWE) calculated using the Chang et al. (1987) algorithm. It is a useful extension to previous studies. However, there are several caveats in the method used for snowmelt onset/end detection in this study as described below.

Specific Comments

First of all, there are no in situ observations used for either thresholds calibration or results validation. I wonder how the authors know the selected thresholds are associated with the actual snowmelt onset/end? Although a manual control dataset was generated and used for results evaluation, the control dataset was produced subjectively from the interpretation of the satellite data only.

The annual peak value of XPGR was used to identify snowmelt onset, which appears to correspond to dates of annual maximum SWE and very low Tb37v (Fig.3). Fig.3 shows that the brightness temperatures at 37GHz are <= 225K on the detected snowmelt onset dates for three out of the four winters. From my experience, the snowpack is unlikely to be melting under such low brightness temperatures. The magnitude of Tb37v was used as a condition for snowmelt detection in some previous studies, but it was always >= 248K.

The XPGR algorithm was developed and used for melt detection on the Greenland ice sheet. Several studies have shown that the XPGR technique detects less melt extent and duration than other algorithms (e.g. Ascraft and Long, 2006; Fettweis et al., 2006). Ascraft and Long (2006) compared six different melt detection methods using either active or passive microwave satellite data over the Greenland ice sheet for the year 2000, including the XPGR method. They found that compared to other methods, XPGR detected significant less melt extent and duration on Greenland (see Table 2 from their paper). Therefore in my opinion the XPGR method is not suitable for

snowmelt detection, especially without proper calibration/validation.

"To determine the end of the snowmelt season, we choose either the date of the yearly maximum Tb37V value, which corresponds to the thinnest snowpack or to a 'bare earth' signal, or the first date where 4 out of 5 days have been within 2 cm of the yearly SWE minimum." SWE calculated using the Chang et al. (1987) algorithm was used for snowmelt end detection in this study. However, the Chang et al. (1987) algorithm was found to overestimate SWE or snow depth in western China (Chang and others, 1992; Che et al., 2008). SWE retrieval from passive microwave data is based on volume scattering of the microwave signals by snow, thus SWE can't be estimated accurately when the snowpack is wet. Most SWE retrieval algorithms are only applied to data from the morning orbit to mitigate the impact of wet snow. The snowpack is likely to be wet and shallow near the end of snowmelt, which would lead to erroneous SWE retrievals from the passive microwave data. Wang et al. (2013) showed that it was a challenge to discriminate wet snow from snow-free land using satellite data alone.

The estimated mean snowmelt periods in the current study are nearly 150 days for large areas of Tibetan Plateau during the 1987-2016 period (Fig.4), while Ke et al. (2016) showed that the annual mean snow cover days were  120 days during the 1981/82 – 2009/10 period based on observations from weather stations (see their Fig.3). This suggests that the detected snowmelt end dates in the current study are likely too late (Fig.5).

On account of the above, I recommend rejection of the paper.

References cited: Ashcraft, I. S., Long, D. G. Comparison of methods for melt detection over Greenland using active and passive microwave measurements. International Journal of Remote Sensing. 2006, 27 (12): 2469-2488.

Chang, A.T.C., J.L. Foster, D.K. Hall, D.A. Robinson, P. Li and M. Cao. 1992. The use of microwave radiometer data for characterizing snow storage in western China. Ann. Glaciol.,16, 215–219.
Che, T., X. Li, R. JIN, R. Armstrong, T. Zhang, 2008, Snow depth derived from passive microwave remote-sensing data in China, Annals of Glaciology, 49, 145-154.

Fettweis, X.; Gallee, H.; Lefebre, F.; van Ypersele, J. P. The 1988-2003 Greenland ice sheet melt extent using passive microwave satellite data and a regional climate model. Climate Dynamics. 2006, 27 (5): 531-541.

Ke et al., 2016, Variability in snow cover phenology in China from 1952 to 2010, Hydrol. Earth Syst. Sci., 20, 755–770, doi: 10.5194/hess-20-755-2016.

Wang, L., Derksen, C., and Brown, R., and Markus, T., 2013, Recent changes in pan-Arctic melt onset from satellite passive microwave measurements, Geophys. Res. Lett., 40, 522–528, doi:10.1002/grl.50098.

---

## Author Comment (AC2) · 13 Jul 2017

Please see attached PDF file. As many comments from both reviewers overlapped, we have included both sets of replies in the attached document.

Please also note the supplement to this comment:
https://www.the-cryosphere-discuss.net/tc-2017-67/tc-2017-67-AC2-supplement.pdf
* * *

---

## Author Response (AR1)

**Reply to Reviewers – Smith et al., 2017**

**Comments - Editor**

*1. Your groundtruthing is not that solid. There is a well-established link between surface air temperatures and melt onset (e.g. Libo Wang et al. 2008, RSE) that you could investigate using surface climate observations. River flow data is another potential source of groundtruth information.*

We have added an external validation section to the MS leveraging MODIS snow covered fraction (MOD10C1, Hall and Riggs, 2016) for assessing the end dates and High Asia Refined Analysis surface temperature (HAR - Surface Temperature, Maussion et al., 2014) for assessing the onset of melt. This is explained in more detail in specific reviewer comments below (General comments of reviewer 1, specific comment 2 of reviewer 2) as well as in the MS.

*2. I suggest you make the study rationale a bit clearer in the Introduction e.g. the reason for monitoring snow melt duration. This is discussed in the Hydrologic Implications section, but it would be useful to make this clear at the beginning of the paper.*

We have updated the introduction to expand on the rationale behind our study.

**Comments – Reviewer 1**

*In this paper, the authors use passive microwave data to identify snowmelt onset, snowmelt end and snowmelt period over the High Mountain Asia (HMA) region. They verify the results of an automated algorithm by comparison to manually identified dates in the microwave signal and find it matches to within 5 days. They then use the algorithm to calculate the melt onset, end and period over 29 years and evaluate trends across the region. The paper is well written and provides a long-term record of snowmelt trends across a region where snowmelt is a critical source of water supply. They use an existing method for identifying melt, but apply new techniques for detecting onset and end, as well as a hierarchical clustering method to identify spatial patterns in the data. This paper contributes to the literature in an understudied area of*
*the world.*

*My main feedback is that the lack of validation data for this technique raises a number of questions. It would be useful to see validation of the approach in this region that would lend confidence to the results, independent of the microwave data. Some possible data sources that could be used include snow covered area from MODIS or VIIRS to estimate snowmelt end dates. Discharge data, if available, could be used to verify the onset of melt by evaluating the rising limb of the snowmelt hydrograph. Similarly, it may be possible to examine shifts towards earlier melt timing by looking at the hydrograph centroid. (See Regonda et al. 2004, Seasonal Cycle Shifts in Hydroclimatology over the Western United States, Journal of Climate, Vol. 18). Alternatively, temperature data may provide some verification of onset dates. If these data are not available, then demonstrating the approach in an area with data would be useful.*

Thank you for the detailed comments and review. We agree that the lack of spatially and temporally extensive control data is a limitation of the study. Unfortunately, long-term, spatially extensive, and high-quality snow records are simply not available for the study region. In our revision, we leverage MODIS MOD10C fractional snow cover (Hall and Riggs, 2016) to assess the reliability of our snowmelt end dates, and High Asia Refined Analysis (HAR) modeled surface temperatures (Maussion et al., 2014) to examine our snowmelt onset dates.

We find that our snowmelt end dates agree very well (slope = 1.00, $R^2$ = 0.99, n=34,468 over 16 years) with the date of MODIS snow clearance (defined here as 5 out of 7 days below 5% snow-covered area). A comparison of MODIS and algorithm snow clearance dates is found below in Figure 1 of this reply. This figure and related discussion have been added to the manuscript in the Results section.

[Figure]

Figure 1 – Comparison of snowmelt clearance dates with MODIS MOD10C1. There is nearly 1-1 agreement between the chosen dates for snowmelt end (slope = 1.00, rsq =0.99, n=34,468). Nearly all algorithm-derived melt dates show less than 5% MODIS fractional snow cover, with nearly 50% showing less than 1% snow cover.

We only compare those dates where there is no cloud cover within 7 days of our algorithm-determined snowmelt end date to limit our analysis to only those years where both methods should provide equally accurate snowmelt end dates.

The accuracy of snowmelt onset dates is somewhat more complicated to determine. Snowmelt onset for this paper could be defined as either the first appearance of liquid water in the snowpack or the beginning of the primary 'melt-off' phase, where there are no more significant snowfall events detected in the passive microwave data and SWE generally starts to decline. In the original manuscript, we chose

simply the highest XPGR peak, which could correspond to either of these dates, depending on the climate – and particularly temperature – context of any given location. In this revision, we have updated the algorithm to flag years where there are multiple coherent peaks in the XPGR time series as being unreliable. These data points are no longer included in the trend analysis components of the paper to ensure that we only examine trends in reliably identified snowmelt dates. We find that ~25% of all snowmelt onset dates across all locations are flagged as unreliable, and thus we use 15% less data overall. Trend values are changed in some cases (cf. Fig 9 in the updated MS), mostly on the Tibetan Plateau where fewer trends are now statistically significant. A small area of the far eastern Tibetan Plateau shifts from slightly positive onset to slightly negative onset as well. Otherwise, the large-scale trend patterns are maintained. A more detailed discussion of this update is given below in response to reviewer 2 - Comment #3, as well as in Figure 2 of this reply.

General comments:

*1. The 36 GHz signal saturates out in deep snow, which I expect much of this area experiences. How does that affect the gradient ratio approach, since the difference may remain fairly constant for much of the season? How do you know you're selecting the actual maximum XPGR?*

Signal saturation is a problem in HMA and elsewhere – particularly in mountainous regions – and dramatically reduces the reliability of SWE estimates. However, even with signal saturation, the appearance of liquid water in the snowpack will cause 'spikes' in the XGPR signal due to the drastic difference between the passive microwave signal response to dry snow (volume scattering of the bare-earth passive microwave signal inside the snowpack) and wet snow (emission of passive microwave signal from the water layer). Our algorithm is sensitive to these spikes and we thus argue that it identifies snowmelt, even in cases where saturation may occur.

*2. Related to question 1, the XPGR seems to follow the calculated SWE signal. How does the calculated SWE compare to general estimates of SWE in the region? Is it reasonable, or is there evidence of signal saturation?*

Unfortunately, in-situ estimates of SWE in the region are sparse. Previous work has reported reasonable results for Western China (Che et al. 2008), but SWE estimates in complex and deep-snow terrain are generally considered unreliable (Tedesco et al., 2015). Without reliable in-situ estimations it is hard to quantify the degree of SWE underestimation due to saturation, but comparisons to modeled results (WRF, HAR) indicate that SWE is underestimated in HMA in some areas.

We aim to exploit shifts native to the time-series of each individual pixel instead of using absolute thresholds for tracking melt, as has been done in previous studies (e.g., Abdalati et al. 1995, Monahan and Ramage, 2010), so the reliability of absolute SWE measurements should not have an outsized impact upon our algorithm.

*3. It is interesting that some of the trends change after 2002, when several additional instruments begin to be available and are included in the analysis. Is it possible that differences in the sensors are causing different results?*

While it is possible that the differences in the sensors could have impacted the results, we think this is unlikely, as we explicitly designed the algorithm to be sensor-independent. First, our results rely on the XPGR, which is a normalized ratio and which should mitigate some, if not all, of the sensor-related differences between instruments. Second, we examine peaks native to each single-instrument time series, so detected melt onset and end dates are native to each instrument. Third, and most importantly, when there are multiple instruments providing a melt date for a given year, we use a conservative strategy for choosing a single melt onset/end date for each year.

For melt onset, if the time series has multiple coherent peaks (defined as two peaks within 5% of each other more than 3 weeks apart, the year is flagged as unreliable for melt onset determination (~25% of the total number of melt onset dates across all locations and years are flagged as such, with a few regions). If two instruments are providing a melt date, and they are less than two weeks apart, we choose the earliest date between the two. If there are three melt onset dates, we choose the median onset date. If the melt onset dates are more than two weeks apart, we flag the year as unreliable. We then use a similar strategy for the end of the melt season, which is generally better constrained by our algorithm. The only difference between the strategy for melt onset and melt end is when there are two end dates more than two weeks apart, we choose the date that is closest to the long-term average melt end date instead of flagging that year as unreliable.

*4. Following on question 3, in section 2.3 the method used to merge the datasets for the hierarchical clustering analysis is described. Was this merged dataset also used in the snowmelt tracking analysis? If not, then explain why differences in the sensors wouldn't impact the estimated melt onset and end dates. If yes then this description should be included earlier.*

The merged dataset was not used for the snowmelt tracking analysis to limit the impact of inter-sensor differences on the determination of melt onset and end. Merging the multiple passive microwave datasets introduces noise to the time series, which in turn impacts placement and magnitude of the peaks used to determine melt onset. We instead used the strategy described in reply to comment #3 above to choose the melt dates for years with multiple sensors. We only use the merged dataset for (1) display purposes on figures in the manuscript, and (2) for the hierarchical clustering. We chose to use the merged dataset for the hierarchical clustering to extend the time period we cluster over and to include data from each sensor to increase our cluster robustness throughout the entire study time period, despite the increase in noise from using multiple merged time series.

*5. The manual selection of dates based on the time series seems subjective. It would be useful to include additional information on how those dates were selected. For example in Figure 3 – in both 2009 and in 2010 there were two peaks of similar magnitude during the winter season. In 2009 the one closer to the end of the season was selected despite appearing less than the earlier one. In 2010 the one very early in the season was selected despite there being an almost equal peak later on. The description in section 3.1 should be clearer.*

We examined not only the XPGR, but the SWE and Tb37V signal at each year to determine snowmelt onset. Both 2009 and 2010 are complex cases, where there are multiple strong candidates for the onset of snowmelt. In 2009, the higher SWE total in the earlier peak, followed by a decreasing by oscillating SWE total, pointed to the earlier XPGR peak as the onset of melt. In 2010, the generally decreasing SWE

total throughout the winter, punctuated one large event and then a return to the previous decreasing curve a few weeks later indicated that the main snowmelt season started earlier. However, in both cases it could be argued that either peak represents the true start of the snowmelt season.

Please see the reply to comment #3 above about changes in the algorithm related to flagging poorly constrained years. Our updated comparison of our algorithm dataset and our control dataset does not include comparison of these poorly constrained melt onset dates. These dates are also no longer used in the assessment of snowmelt trends.

Specific comments:
*1. Page 3, Line 25: why was this algorithm chosen over the other methods referenced in the introduction?*

We choose this method due to (1) simplicity, (2) speed of calculation, and (3) lack of reliance on pre-calculated metrics or assumptions. For example, diurnal temperature algorithms rely on fixed differences between day and night temperatures to detect melt; these differences are neither constant in space nor in time (intra- and inter-seasonal) across our large and diverse study area. We found that the XPGR algorithm was well-suited to the time series approach we use, and was fast enough to compute for the entire dataset we used. It has the additional advantage of only relying on night-time data, which somewhat limits the impact of sporadic daytime melt (due to solar radiation) on our results.

*2. Section 2.1: additional background information on passive microwave detection of snow and snowmelt is needed, specifically on how the signal is affected by liquid water in the snow at different frequencies.*

Section 2.1 (Section 2.2 in the new MS) has been updated with additional information on the interactions of snow and passive microwave radiation.

*3. Page 4, Line 7: Was SWE calculated using the Chang algorithm on Tb from the different sensors? Or are you using the SWE products developed for the different sensors? Adding the equation would be useful. How do you combine multiple sensors when available?*

We use the original Chang algorithm for each sensor, albeit tuned to each sensor (i.e., the Chang algorithm is offset by 5K when the AMSR-E frequencies are used). As our algorithm relies primarily on the normalized XPGR index and we do not aim to provide a tightly-constrained SWE product, we leverage a single, consistent algorithm across all satellite datasets. We have added the equation to the MS in Section 2.1. We combine the SWE datasets for display purposes (e.g., Figure 3 of the MS) by resampling a combined, multi-instrument, timeseries to the daily mean SWE value.

*4. Page 5, line 2: Not sure what is meant by "regularize".*

We perform a simple linear regression of the overlapping pieces of the time series to determine the offset between the overlapping datasets. We then add or subtract the determined coefficients to bring

the time series closer to a single coherent dataset (cf. Figure 2 of the MS) in order to use the longest possible time frame for our hierarchical clustering. We have clarified this in-text.

*5. Page 5, lines 23-25: How does the standard deviation of the melt onset date vary spatially? It seems this approach would work best in high elevation/deep snow regions, whereas along the edges in lower elevation where the snow is more ephemeral there might be more error. This would also affect estimates of melt period.*

Very true. However, we find that the highest deviation regions are those areas where deep snow impacts the algorithm, or late-season storms change the snowmelt onset date from year-to-year. Ephemeral snowmelt is actually tracked quite well, as it tends to have sharp peaks and a single, continuous melt-off curve. Those areas where snow varies significantly year-to-year, and areas with multiple SWE peaks, are more difficult (ie, in the Karakoram, cf. Fig 2 of this reply and Figure S5 in the revised manuscript). We have attempted to reduce this error by flagging unreliable years in our melt dataset as described above. A map of the standard deviation of melt onset dates for 29 years has also been added to the Supplement (Figure S5). Note that many low-STD regions are areas where there is almost no snow (e.g., Tarim Basin and low-elevation areas of the Himalayan front). These areas are not included in our trend analysis.

[Figure]

Figure 2 – Standard deviation in snowmelt period, calculated over the full 29-year dataset. Low standard deviation areas follow low-SWE and low-elevation zones, with more variable snowfall regions (ie, Karakoram) showing higher standard deviations.

*6. Page 6, line 8: Where you say, "snow is present for less than a month on average." Are you referring to the snowmelt period or the entire snow season? That sounds like the entire season, but everywhere else is referring to melt period.*

You're correct, we refer here to short snowmelt periods. We remove data that have a long-term average snowmelt period of less than 20 days. We have updated the wording in the MS.

*7. Page 6, line 24: You say, "As can be seen in Figure 3, inter-annual variation in snowfall can cause large disparities in the yearly dates of snowmelt onset and end." Based on Figure 3 there doesn't appear to be a lot of variability – the peak SWE is around 100mm each year. Are you referring to timing of snowfall events?*

Yes, we refer here to the timing and magnitude of SWE buildup and melt. Each year has quite different peak locations, and relative peak sizes. This has been clarified in the MS.

*8. Figure 1. Can you identify on the overview map the location of the sample data shown in figures 2 and 3?*

This has been updated.

*9. Figure 4: What do the gray areas on the plateau represent? Provide an explanation, similar to figure 6.*

This has been updated.

**Comments – Reviewer 2**

*Passive microwave satellite data have been used for snowmelt onset detection on nearly all components of the cryosphere (e.g. ice sheets, sea ice, lake ice, and seasonal snow cover) mainly at the northern middle to high latitudes and Antarctica, where there is permanent or relatively stable snow accumulation each winter. It has also been used to classify daily freeze/thaw state dynamics for global vegetated land cover regions without distinguishing individual elements of the landscape (e.g. soil, vegetation, snow). The current study detects snowmelt onset/end dates and snowmelt season in Tibetan Plateau and its surrounding mountain ranges, and analyzes trends in snowmelt timing during the 1987-2016 period. Snowmelt onset detection is based on the crossed-polarized gradient ratio (XPGR) algorithm developed in previous studies, and the end of snowmelt is determined by time series of Tb37v and snow water equivalent (SWE) calculated using the Chang et al. (1987) algorithm. It is a useful extension to previous studies. However, there are several caveats in the method used for snowmelt onset/end detection in this study as described below.*

Specific Comments

*First of all, there are no in situ observations used for either thresholds calibration or results validation. I wonder how the authors know the selected thresholds are associated with the actual snowmelt onset/end? Although a manual control dataset was generated and used for results evaluation, the control dataset was produced subjectively from the interpretation of the satellite data only.*

In the updated manuscript we have added a section comparing our results to MODIS snow cover and HAR mean daily surface temperature (see also reply to comment #1 of reviewer 1, and the extended discussion of comparison to control datasets in response to your comment below). In situ data at the correct spatial and temporal scale simply does not exist for the majority of the study region, so we rely on these proxies instead. An updated discussion of the caveats of the method and uncertainties is included in the MS.

*The annual peak value of XPGR was used to identify snowmelt onset, which appears to correspond to dates of annual maximum SWE and very low Tb37v (Fig.3). Fig.3 shows that the brightness temperatures at 37GHz are <= 225K on the detected snowmelt onset dates for three out of the four winters. From my experience, the snowpack is unlikely to be melting under such low brightness temperatures. The magnitude of Tb37v was used as a condition for snowmelt detection in some previous studies, but it was always >= 248K.*

Thank you for this important critique. We have modified our algorithm as described in comment #3 of reviewer #1. In addition to this, we have compared our melt onset dates to both MODIS fractional snow-covered area and HAR modeled surface temperature. Figure 3 below shows the HAR surface temperature distribution at melt onset. While average daily temperatures tend to be negative, daytime temperatures are often positive (average 4C), with large (20+ degree) temperature variation. Figure 4 below shows a direct comparison between our melt onset dates and the MODIS and HAR datasets. As can be seen in the middle panel, the onset of melt correlates with the peak of MODIS fractional snow cover, and with the yearly minimum temperature from HAR (this has been added to the Supplement as Figure S2). This implies that our melt algorithm is capturing the turning point where snow ceases to increase and starts melting out. These figures have been added to the Supplement.

We also compared HAR air temperature to the raw Brightness Temperature (Tb) values (Figures 5 and 6 of this reply and S3-4 in the updated Supplement), and found that while there is a definite correlation between air temperature and Tb, positive surface temperatures are associated with a wide range of possible Tb values. The Tb-Temperature distribution is also quite different between spatial locations, implying that a single threshold would not be appropriate for determining the possibility of snowmelt from Tb values.

[Figure]

Figure 3 – HAR modeled hourly surface temperature (Maussion et al., 2014) at the date of algorithm-melt onset. Average daily temperature (red) and average daytime temperature (blue) show divergent means, where the onset of melt is characterized by below zero average temperatures, but daytime temperatures that are positive. The average daily range of temperatures (black) shows quite large variability in 24h temperature profile at the onset of snowmelt.

[Figure]

Figure 4 – Representative sample point showing SWE (top, this study), MODIS fractional snow covered area (middle, Hall and Riggs, 2016), and HAR daily average temperature (bottom, Maussion et al., 2014) over the period 2001-2009. Algorithm-derived melt onset dates (dashed lines, black) and end dates (solid lines, red). Years with multi-peaked XPGR data do not return a melt onset date. Melt onset dates correlate well with peak annual snow-covered area (middle) and the yearly minimum temperature (bottom), implying that our melt algorithm captures the onset of the snowmelt season accurately. Data taken from 71.25E, 36.75N.

[Figure]

Figure 5 – HAR Average daily temperature vs Temperature Brightness (37V in green, 18H in red). Both channels show correlations with air temperature, but show a wide spread. This observation indicates that there is no single Temperature Brightness threshold that can be used for snowmelt detection.

[Figure]

Figure 6 – HAR temperature metrics vs Tb at snowmelt onset. Both 37V (blue) and 18H (red) channels show significant spread. While there is a slight correlation between average daily temperature and Tb, average daytime temperature is very poorly related to Tb. This implies that the night-time passive microwave data we use to track snowmelt onset still captures the impacts of above-zero daytime temperatures.

*The XPGR algorithm was developed and used for melt detection on the Greenland ice sheet. Several studies have shown that the XPGR technique detects less melt extent and duration than other algorithms (e.g. Ashcroft and Long, 2006; Fettweis et al., 2006). Ascraft and Long (2006) compared six different melt detection methods using either active or passive microwave satellite data over the Greenland ice sheet for the year 2000, including the XPGR method. They found that compared to other methods, XPGR detected significant less melt extent and duration on Greenland (see Table 2 from their paper). Therefore, in my opinion the XPGR method is not suitable for snowmelt detection, especially without proper calibration/validation.*

In our usage of the XPGR, we do not rely on static melt thresholds, as has been done in the original Abdalati and Steffen (1995) study and the studies of Fettweis et al. (2006) and Ashcroft and Long (2006). Ashcroft and Long (2006) note that "with XPGR, local maxima in q(t) occur at times similar to those observed for the a-based and Tb-M methods; however, the relative amplitude of the peaks are different, contributing to discrepancies in the melt detection by XPGR and the other methods." This implies that the timing of melt events is detected in the XPGR time series, but the use of a single threshold calculated for all of Greenland negatively impacts melt detection. High elevation and internal areas are classified with the same cutoff as low elevation and coastal areas in their study. In our method, we do not rely on a single cutoff, but instead find the XPGR peak unique to individual years and locations.

It should also be noted that we do not classify individual days as melting or not melting, as has been done in the previously mentioned studies, but instead seek to identify the primary start and end dates of the snowmelt season. Ashcroft and Long (2006) further note "the differences in the melt detected by the individual methods are attributed to differences in sensitivity to melt due to frequency and/or differences in the definition of melt implicit with each method." Based on our stated goal of identifying the primary melt period, and the data comparisons presented in Figures 1-6 of this reply, we maintain that the XPGR is a useful metric.

*"To determine the end of the snowmelt season, we choose either the date of the yearly maximum Tb37V value, which corresponds to the thinnest snowpack or to a 'bare earth' signal, or the first date where 4 out of 5 days have been within 2 cm of the yearly SWE minimum."* SWE calculated using the Chang et al. (1987) algorithm was used for snowmelt end detection in this study. However, the Chang et al. (1987) algorithm was found to overestimate SWE or snow depth in western China (Chang and others, 1992; Che et al., 2008). SWE retrieval from passive microwave data is based on volume scattering of the microwave signals by snow, thus SWE can't be estimated accurately when the snowpack is wet. Most SWE retrieval algorithms are only applied to data from the morning orbit to mitigate the impact of wet snow. The snowpack is likely to be wet and shallow near the end of snowmelt, which would lead to erroneous SWE retrievals from the passive microwave data. Wang et al. (2013) showed that it was a challenge to discriminate wet snow from snow-free land using satellite data alone.*

We realize the shortcomings of the Chang algorithm, particularly for wet snow depth estimation at the end of the snowmelt season. This is why we do not simply identify snow clearance as days where SWE reaches zero, but allow the algorithm to declare the end of the melt season when small amounts of SWE remain, or when Tb37 has reached its yearly max. In our now included validation with MOD10C, we find very close agreement between our melt end dates and the MOD10C snow fraction dropping below 5% for five out of seven consecutive cloud-free days (Figure 1 of this reply, Figure 4 in the updated MS). We thus argue that our method successfully identifies the end of the snowmelt season despite problems inherent with passive microwave SWE estimation.

*The estimated mean snowmelt periods in the current study are nearly 150 days for large areas of Tibetan Plateau during the 1987-2016 period (Fig.4), while Ke et al. (2016) showed that the annual mean snow cover days were 120 days during the 1981/82 – 2009/10 period based on observations from weather stations (see their Fig.3). This suggests that the detected snowmelt end dates in the current study are likely too late (Fig.5). On account of the above, I recommend rejection of the paper.*

The reviewer raises an important point that we address with the following four comments:

(1) The cited Ke et al. (2016) study uses local station data to estimate the length of the snow-cover season for Western China. However, from their Figure 1, their station density is very poor, particularly in the Tibetan interior. Their following Figure 3 (snow-covered days) seems to be interpolated from these sparse points, which may or may not be representative of the large, unmonitored areas. Additionally, other studies of snowcover in Tibet (e.g., Pu et al., 2007) note extensive areas of 9+ month snow cover using MODIS data (their Figure 4).

   The coherence between independent snow-clearance measures (Figure 1 of this reply, MODIS and Passive Microwave) leads us to argue for the reliability of our snowmelt clearance dates. The non-representative spatial and elevation distribution of weather stations in the Ke et al. (2016) study, along with difficulties of scaling up snow measurements at point locations and interpolating them over diverse terrain, could account for some of the difference between the results of the two studies.

(2) As snowmelt period depends on two measurements (the onset and the end date), discrepancies between our snowmelt periods and those shown in Ke et al. (2016) are more likely to come from mismatches in the snowmelt onset date. However, as is shown in Figures 2 and 3 of this reply, our melt onset dates generally track the start of the upward arm of the yearly temperature distribution, and are correlated with dates of positive daytime surface temperatures.

(3) In order to examine the reviewer's comments about snowmelt period in more detail, we tested an additional change to the melt tracking algorithm with an imposed Tb threshold on the onset date of melt (after Fettweis et al., 2006). This threshold was chosen so that snowmelt was only found if the Tb18H was above the long-term average Tb18H plus one half standard deviation. This method was used previously in Greenland to improve the XPGR method with the stated goal of flagging each day of the year as melting or not melting.

   We find that this dramatically reduces snowmelt periods (Fig. 7 of this reply, below), but also reduces the agreement between our algorithm data and the MODIS and HAR control datasets. Onset dates correlate with quite low (~10%) snow fraction, and with positive (+15C) average daily temperatures. Melt periods in regions outside of the Tibetan Plateau shrink significantly, and no longer match up with other published literature. This implies that by imposing this threshold, we examine only the very end of the snowmelt season, and not the true onset of snowmelt.

   Importantly, we also compared trends in our data using both methods (adaptive XPGR threshold and with the addition of a fixed Tb threshold) and found that most regions maintained the same trend direction, if not trend magnitude. Thus, trends in melt end date and melt period are similar between both methods. The largest discrepancy is in the far eastern Tibetan Plateau, where melt onset trends shift from negative to positive when the fixed thresholding method is

used. The northern regions of HMA (north of the Tien Shan) also do not exhibit statistically significant trends for all three metrics (onset, end, period). We attribute these differences to the fact that the two methods are really tracking two separate measures of snowmelt onset. In our method (which is used and described in the MS), we track the turning point of the yearly temperature cycle and the peak time when snow first starts to decrease. Using the threshold metric, we track the last strong pulse of snowmelt, when much of the snow has already melted or sublimated away. For these reasons, we maintain that our algorithm, with the changes described here in this reply and in the updated MS, is a valid way to track the snowmelt in this context.

(4) Lastly, we argue that the context for the previously cited snowmelt studies and this study are different. Most of the snowmelt algorithms have been developed over Greenland or for tracking sea ice. As the altitudes and latitudes of these study locations are drastically different, solar radiative forcing is generally larger in High Mountain Asia. Additionally, we modify the static approach used in previous algorithms to derive snowmelt over a wide range of topographies, and thus the efficacy of the previous snowmelt algorithms and our algorithm cannot be directly compared.

[Figure]

Figure 7 – Average snowmelt period when a more conservative Tb18H threshold is applied to the method. While melt periods in Tibet match better with those proposed in Ke et al. (2016), melt periods in the western regions of HMA are unreasonably low.

We argue that despite the shortcomings of passive microwave data – and the lack of large-scale ground control data – our algorithm accurately tracks the onset and end of the snowmelt season. An updated discussion of methods, caveats, and comparisons to the control datasets have been added throughout the manuscript.

[revised manuscript text omitted]

---

## Referee Report (RR1)

Revised manuscript review: The Cryosphere

Title: Spatio-temporal Patterns of High Mountain Asia's Snowmelt Season Identified with an Automated Snowmelt Detection Algorithm, 1987-2016

Taylor Smith1, Bodo Bookhagen1, and Aljoscha Rheinwalt1

1Institute for Earth and Environmental Sciences, Universität Potsdam, Germany

Correspondence to: Taylor Smith (tasmith@uni-potsdam.de)

Comments:

In the revised paper, the authors included some comparisons between their results and other datasets (MODIS fractional snow cover and HAR surface temperature), which is helpful. However, it appears that the derived snowmelt onset dates in the paper are not associated with actual snow melting, instead as stated in the response: "Figure 4 below shows a direct comparison between our melt onset dates and the MODIS and HAR datasets. As can be seen in the middle panel, the onset of melt correlates with the peak of MODIS fractional snow cover, and with the yearly minimum temperature from HAR (this has been added to the Supplement asFigure S2). This implies that our melt algorithm is capturing the turning point where snow ceases to increase and starts melting out. These figures have been added to the Supplement."

Figure 5 also shows that at the detected snowmelt onset dates, daytime mean temperature ranges from -30 to 30ºC, thus many cases with temperatures below the freezing point.

Thus snowmelt onset date from this study is not the same as in previous studies, which is usually the date associated with the appearance of liquid water in snow and near freezing point surface temperatures.

In addition, I found the temporal variations of XPGR in Fig.3 of this paper over seasonal snow cover are different from those for permanent snow on Greenland in previous studies. For example, in Fig.2 of Abdalati and Steffen (1995), large XPGR values are concentrated during the summer melt period, with much lower values during the frozen period. In Fig.3 of this paper, XPGR increases gradually from the beginning of the year until reaching its annual peak around the date of maximum SWE, then decreases to the annual minimum in the summer, with secondary peaks of ups and downs in between. This warranties careful calibration/validation of using the XPGR method in HMA region.

For curiosity and to fulfill my responsibility as a reviewer, I plotted (shown below) the daily XPGR (red, from SSM/I EASE-grid 25km data), daily mean surface air temperature (green), and daily snow depth (blue) at Tarko-Sale weather station (64.917ºN, 77.817ºE), unfortunately not in HMA. The time series of XPGR in my plot exhibits similar temporal variations as shown in Fig.3 of the current paper, which is likely the case for season snow cover. The maximum XPGR (~0.06) occurred on DOY66, with air T of -

22.9°C. The air T did rise to near the freezing point on DOY70 (-0.4 °C), however, snow depth didn't show much decrease until after DOY127.

[Figure]

The above plot suggests that dates with maximum XPGR do not correspond to snowmelt onset, at least not the commonly defined snowmelt onset as in previous studies. To avoid misleading, I'd suggest the authors use another term such as maximum SWE instead of snowmelt onset, and also justify the applications of their results accordingly. They could also compare their results with those from a recent publication on snowmelt detection over HMA:

Chuan Xiong, Jiancheng Shi, Yurong Cui, and Bin Peng, Snowmelt Pattern Over High-Mountain Asia Detected From Active and Passive Microwave Remote Sensing.
http://ieeexplore.ieee.org/document/7930395/

Xiong et al (2017) indicates that previous melt detection algorithms developed for polar regions may not work well in HMA due to its complex topography. They proposed a method using a combination of median filters and first order derivative for snowmelt detection from active and passive microwave data.

---

## Author Response (AR2)

**Reply to Reviewers – Smith et al., 2017**

**Comments – Reviewer 2**

*__General Comments:__*

*In the revised paper, the authors included some comparisons between their results and other datasets (MODIS fractional snow cover and HAR surface temperature), which is helpful. However, it appears that the derived snowmelt onset dates in the paper are not associated with actual snow melting, instead as stated in the response: "Figure 4 below shows a direct comparison between our melt onset dates and the MODIS and HAR datasets. As can be seen in the middle panel, the onset of melt correlates with the peak of MODIS fractional snow cover, and with the yearly minimum temperature from HAR (this has been added to the Supplement as Figure S2). This implies that our melt algorithm is capturing the turning point where snow ceases to increase and starts melting out. These figures have been added to the Supplement."*

*Figure 5 also shows that at the detected snowmelt onset dates, daytime mean temperature ranges from -30 to 30°C, thus many cases with temperatures below the freezing point. Thus snowmelt onset date from this study is not the same as in previous studies, which is usually the date associated with the appearance of liquid water in snow and near freezing point surface temperatures.*

*In addition, I found the temporal variations of XPGR in Fig.3 of this paper over seasonal snow cover are different from those for permanent snow on Greenland in previous studies. For example, in Fig.2 of Abdalati and Steffen (1995), large XPGR values are concentrated during the summer melt period, with much lower values during the frozen period. In Fig.3 of this paper, XPGR increases gradually from the beginning of the year until reaching its annual peak around the date of maximum SWE, then decreases to the annual minimum in the summer, with secondary peaks of ups and downs in between. This warranties careful calibration/validation of using the XPGR method in HMA region.*

*For curiosity and to fulfill my responsibility as a reviewer, I plotted (shown below) the daily XPGR (red, from SSM/I EASE-grid 25km data), daily mean surface air temperature (green), and daily snow depth (blue) at Tarko-Sale weather station (64.917°N, 77.817°E), unfortunately not in HMA. The time series of XPGR in my plot exhibits similar temporal variations as shown in Fig.3 of the current paper, which is likely the case for season snow cover. The maximum XPGR (~0.06) occurred on DOY66, with air T of -22.9°C. The air T did rise to near the freezing point on DOY70 (-0.4 °C), however, snow depth didn't show much decrease until after DOY127.*

[Figure]

*The above plot suggests that dates with maximum XPGR do not correspond to snowmelt onset, at least not the commonly defined snowmelt onset as in previous studies. To avoid misleading, I'd suggest the authors use another term such as maximum SWE instead of snowmelt onset, and also justify the applications of their results accordingly. They could also compare their results with those from a recent publication on snowmelt detection over HMA:*

*Chuan Xiong, Jiancheng Shi, Yurong Cui, and Bin Peng, Snowmelt Pattern Over High-Mountain Asia Detected From Active and Passive Microwave Remote Sensing.*
*http://ieeexplore.ieee.org/document/7930395/*

*Xiong et al (2017) indicates that previous melt detection algorithms developed for polar regions may not work well in HMA due to its complex topography. They proposed a method using a combination of median filters and first order derivative for snowmelt detection from active and passive microwave data.*

Thank you for this detailed and thoughtful reply.

The premise of the XPGR is that the peaks (or rapid changes) in XPGR are linked to the appearance of liquid water in the snowpack, due to the differences between the passive microwave properties of dry and wet snow. However, the reviewer raises a good point -- we don't have convincing control data in HMA to say whether the XPGR in a seasonal snowmelt situation is really tracking snowmelt onset or is more tied to peaks in SWE, snowpack metamorphosis, or some other factor. Methods developed for tracking snowmelt in polar regions such as Greenland and the Arctic are not directly transferable to HMA and we have attempted to adjust and account for that.

Looking again at the figures from Abdalati and Steffen (1995) where XPGR was first described (Figures. 2-4 of that manuscript) their PM data actually looks pretty similar to ours in that 'warm' regions of Greenland show a similar seasonal oscillation pattern in XPGR – albeit with sharper peaks. Abdalati and Steffen (1995) choose to identify days as either having some liquid water or not based on a simple threshold, and split the year into 'melting' and 'non-melting' parts. As we have snow-free, snow-buildup, and snow-melt parts of the year, we are certainly tracking something a bit different than the snowmelt/freezing cycle over ice that they examine in that paper.

Despite this, our algorithm still tracks a consistent metric that is linked to snowpack character year-over-year for each pixel/location – whether that metric is related to the short appearance of liquid water, the beginning of strong snowpack metamorphism, or simply peak SWE. As such, the trend detection segment of our paper still provides valuable insight into the long-term changes in snow timing in HMA.

To make this clearer in the MS and to avoid confusion, we have changed our terminology to refer to maximum XPGR, which is driven by peak separation between the Tb18 and Tb37 channels. We refer to this throughout as MXPGR, and have added an updated discussion of the reasoning behind using this metric and its utility.

In developing our methodology, we had tried a similar approach to that of Xiong et al. (2017) involving filtered time series and an analysis of slope and breakpoints. We found that the required level of smoothing (Xiong et al. use a 31-day window) washed out the peaks significantly, and made it difficult to determine where within a several day window the original peak actually lay. We also found that using a

single smoothing window across many diverse areas was unreliable due to the drastic differences in the shape of snow buildup and melt curves across HMA. In our tests, we weren't able to find a smoothing function that satisfied our requirements of both clearly maintaining the peak location and strongly smoothing out high-frequency noise.

Despite the differences in our methodology, Xiong et al.'s (2017) pattern of melt onset trends is quite similar to ours (cf. our Figure 9, their Figure 6). Both methods identify negative trends in much of HMA, with positive trends mostly confined to the Karakoram and Kunlun Shan. We identify a slightly different trend in eastern Tibet, and lack statistically significant results for some areas where they identify trends, but maintain the same general spatial pattern. Unfortunately, they do not explain how they choose points that have 'effective snowmelt onset date detection' or how they determined the statistical significance of their results, which makes a direct comparison with their results difficult. We found point-by-point linear regressions unworkable in HMA due to large inter-annual variations in the snowmelt season, and thus relied on a hierarchical clustering approach to assess trends in the timing of the snowmelt season.

**Comments – Reviewer 3**

*This is a well-defined summary of statistical spatio-temporal behavior of snowmelt onset and end in High Mountain Asia, using one of several melt onset algorithms using passive microwave brightness temperatures. The authors clearly indicate the value of predictability of snowmelt runoff for downstream water uses, and conclude that variability in derived regional runoff patterns warrant both regional and small-scale studies for effective water management. They make use of robust and carefully considered methods to generate and analyze the melt onset and end data sets.*

*The comments from the two current reviewers have covered all of my earlier concerns and the response from the authors looks good to me. Glad they followed through with application of MODIS data to compare with "snowmelt end" estimates.*

*Just a few comments from reading through the revised version.*

Thank you for these comments. We address the individual comments below.

*General comments:*

*(p. 2, lines 18-26) The authors enumerate a variety of passive microwave snowmelt algorithm approaches, and then choose to work with the XPGR method of Abdalati and Steffen, changing the original fixed threshold to a more dynamically-determined threshold. I think a short statement about this reasoning would be useful to the reader, why did they choose this method over the others to begin with?*

We choose this method due to (1) simplicity, (2) speed of calculation, and (3) lack of reliance on pre-calculated metrics or assumptions. For example, diurnal temperature algorithms rely on fixed differences between day and night temperatures to detect melt; these differences are neither constant in space nor in time (intra-and inter-seasonal) across our large and diverse study area. We found that the XPGR algorithm was well-suited to the time series approach we use, and was fast enough to compute for the entire dataset we used. It has the additional advantage of only relying on night-time data, which somewhat limits the impact of sporadic daytime melt (due to solar radiation or otherwise) on our results. We have added an additional statement about this in the updated MS.

*(page 4, section 2.2) I believe I understand how the manual control dataset generation was accomplished, but how do the authors know that these control answers are correct at identifying the sought-after signals in reality?*

The lack of spatially and temporally extensive in-situ snowmelt with which to calibrate our algorithm is a significant drawback. Unfortunately, such data simply does not exist, so we base our results on certain

assumptions drawn from the passive microwave time series, modeled air temperatures, and MODIS snow cover data. Reviewer #2 raised a similar issue, and posited that the peak XPGR may not in fact identify the onset of snowmelt as has been defined in previous studies (see more detailed comments above in the response to Reviewer 2). In our updated manuscript, we have modified the language around the onset of snowmelt to reflect the fact that we really are tracking the maximum XPGR, or maximum separation between the Tb18 and Tb37 signals. While this metric tracks snowmelt well in Greenland, as has been established in previous studies, it is unclear whether this remains the case in the complex terrain of HMA.

Despite this, we track a consistent metric between years for each location, and thus trends in the 'maximum XPGR' remain useful for identifying changes in the snowmelt season – whether those changes are the true onset of snowmelt as defined in other studies or simply a consistent metric of maximum passive microwave signal separation.

*(page 4, line 30) The authors state that the XPGR is "not sensitive to SWE calibration issues." I am not sure this statement is justified for the original XPGR algorithm (with the fixed threshold). I believe it is justified for the authors' modified method, which identifies XPGR peaks regardless of numeric values, without referring to an empirical threshold. Perhaps coining a phrase for the modified XPGR technique would clarify this point?*

We have modified this text slightly, in regards to the comments of the reinterpretation of the XPGR maximum.

*(Supplemental material, Table S1) I am unfamiliar with the data citation requirements of this journal. While it is useful to include the input data algorithm versions and processing levels in this table, it would also be useful to include complete data set citations (with DOIs, if available), in the main reference list, to assist in citation-tracking algorithms and to be completely explicit as to input data versions and data sources used.*

This has been added to Table S1.

*(Supplemental material, figure S1, Flowchart of Melt Tracking Algorithm) I find this flowchart difficult to understand, I don't see enough information here for a reader to really understand the logic or flow. While it is called a flowchart, it is not using traditional flowchart symbols for logical direction and conditional statements. Presumably the chart is read from top to bottom, but this is an assumption (without arrows to indicate the direction). There is an extra condition described in the flowchart, based on the 60-day difference of end-of-melt from "long term average" that is not explained or justified in the narrative. Also, how would one know the long-term average before calculating the values for all year? This looks circular to me, I'm sure I'm missing something in the details. Perhaps*

*it means that the top box is executed for all years, first, in order to take an average and proceed down the logic? However, these are all assumptions on my part and it would be better for the authors to be explicit than to require a reader to guess.*

You are correct in your interpretation of the flowchart. We have updated this figure to be clearer, and included the preliminary step of determining long-term average onset and end dates. The long-term average step isn't necessary, but it speeds up the processing by only re-calculating values that are unlikely to be reasonable (i.e., snowmelt onset being detected in December, when the long-term average is in May), and accepting values which are closer to the long-term mean. The full algorithm code (Python) is available in a public repository as well: https://github.com/UP-RS-ESP/SnowmeltTracking.

*(Figures 5 and 8) The polar plots in these figures are clearly rich in multiple dimensions. However, I cannot puzzle out what determines the placement of a value along a given radial line. It could just be my ignorance of this type of plot, but I cannot find an explanation of this part of the dimension space. Do "bin heights" mean "distance from the center"? This confusion on my part makes it more difficult to interpret the plots. This problem could be remedied by a short explanation of radial position in the caption for readers unfamiliar with this plotting convention.*

Bin heights indeed means distance from the center, in this case normalized to the largest bin. Thus, the single largest bin will reach all the way to the edge of the circle, with bins (days) with smaller total numbers of melt onset/end/periods will be shorter. We have updated this explanation in text and refer to this as radial bin heights.

*Grammatical/typographical comments:*

*(p. 1, line 30, p. 3 line 18) I believe "is comprised of" should simply be "comprises." I do admit this objection is arguable and that the former usage is gaining acceptance over time.*

This has been updated.

[revised manuscript text omitted]